# Neonatal and maternal outcomes following SARS-CoV-2 infection and COVID-19 vaccination: a population-based matched cohort study

Understanding the impact of SARS-CoV-2 infection and COVID-19 vaccination in pregnancy on neonatal and maternal outcomes informs clinical decision-making. Here we report a national, population-based, matched cohort study to investigate associations between SARS-CoV-2 infection and, separately, COVID-19 vaccination just before or during pregnancy and the risk of adverse neonatal and maternal outcomes among women in Scotland with a singleton pregnancy ending at ≥20 weeks gestation. Neonatal outcomes are stillbirth, neonatal death, extended perinatal mortality, preterm birth (overall, spontaneous, and provider-initiated), small-for-gestational age, and low Apgar score. Maternal outcomes are admission to critical care or death, venous thromboembolism, hypertensive disorders of pregnancy, and pregnancy-related bleeding. We use conditional logistic regression to derive odds ratios adjusted for socio-demographic and clinical characteristics (aORs). We find that infection is associated with an increased risk of preterm (aOR=1.36, 95% Confidence Interval [CI] = 1.16–1.59) and very preterm birth (aOR = 1.90, 95% CI 1.20–3.02), maternal admission to critical care or death (aOR=1.72, 95% CI = 1.39–2.12), and venous thromboembolism (aOR = 2.53, 95% CI = 1.47–4.35). We find no evidence of increased risk for any of our outcomes following vaccination. These data suggest SARS-CoV-2 infection during pregnancy is associated with adverse neonatal and maternal outcomes, and COVID-19 vaccination remains a safe way for pregnant women to protect themselves and their babies against infection.

SARS-CoV-2 infection during pregnancy has been associated with adverse neonatal and maternal outcomes. There is relatively extensive published evidence for selected pregnancy outcomes, including maternal admission to critical care and preterm birth[1,2]; however, there remains a lack of detail to inform clinical practice. A recent meta-analysis estimated that women with (compared to without) SARS-CoV-2 infection in pregnancy had 60% greater odds of preterm birth[2]. However, poor reporting on whether preterm births were spontaneous (i.e., after spontaneous rupture of membranes or onset of

contractions) or provider-initiated (i.e., following induction of labour or pre-labour caesarean section) precluded disentangling the relative importance of these different causes. The evidence for association between infection and some outcomes, including pregnancy-related hypertension and bleeding, is weak or conflicting, and there are very few studies for any outcomes that explore whether gestation at infection modifies the outcome risk. These evidence gaps mean that there is ongoing uncertainty regarding the antenatal and intrapartum care of women who have SARS-CoV-2 in pregnancy. For example, the

e-mail: Rachael.Wood@phs.scot

need for preterm birth risk assessment[3], enhanced screening for pre-eclampsia, additional antenatal ultrasound growth surveillance, or continuous foetal monitoring in labour all remain contentious and therefore absent from guidelines for managing SARS-CoV-2 infection in pregnancy[4,5].

A growing body of evidence suggests COVID-19 vaccination is safe for pregnant women, as well as an effective way to reduce the risks associated with SARS-CoV-2 infection for both them and their babies. Maternal COVID-19 vaccination has not been associated with adverse pregnancy-related outcomes[6–11] and has been found to be effective in reducing the risk of SARS-CoV-2 infection and associated adverse outcomes[6,12,13]. Despite this growing body of evidence on both the safety and effectiveness of COVID-19 vaccination in pregnancy, vaccination uptake has been relatively low among pregnant women, with concerns about safety primarily driving this vaccine hesitancy[6,14].

Helping pregnant women and their healthcare providers make informed decisions on the importance of COVID-19 vaccination in pregnancy requires high-quality data on: (1) the impact of SARS-CoV-2 infection (in the absence of vaccination) on adverse maternal and neonatal outcomes; (2) the safety of vaccination in pregnancy; and (3) the effectiveness of vaccination in reducing the impact of the infection (either by preventing infection or reducing the severity of infection). In our study, we examined the first two of these three issues using data from the COVID-19 in Pregnancy in Scotland (COPS) database[15,16]. Specifically, we used national, population-based data from Scotland to create matched cohorts to investigate whether there was an association between SARS-CoV-2 infection and, separately, COVID-19 vaccination in the six weeks preconception or during pregnancy and the subsequent risk of adverse neonatal (stillbirth, neonatal death, extended perinatal mortality, low Apgar score [5 min Apgar score <7], very low Apgar score [5 min Apgar score <4], small-for-gestational age [birthweight <10th centile], very small-for-gestational age [birthweight <3rd centile], preterm birth [<37 weeks gestation] and very preterm birth [<32 weeks gestation]) or maternal outcomes (admission to critical [intensive or high dependency] care [any cause] or death [any cause], hypertensive disorders of pregnancy, pregnancy-related bleeding, and venous thromboembolism).

## Results

### SARS-CoV-2 infection and neonatal outcomes
The COPS study database contained information on 81,441 singleton pregnancies that reached at least 20 weeks, 0 days (20 + 0) gestation, ended in a live or stillbirth, and where the pregnancy was ongoing or conceived after the start of widespread community testing for SARS-CoV-2 infection on May 18, 2020, and conceived before June 2, 2021. Of these, 4074 had confirmed infection (and no vaccination) between six weeks preconception up to the end of pregnancy. 1429 had both infection and vaccination during the pregnancy exposure period and were excluded. An additional 11,379 had vaccination during the pregnancy exposure period, leaving a pool of 64,559 uninfected (and unvaccinated) control pregnancies for matching to the infected pregnancies on maternal age at conception, season of conception and gestational week at exposure/matching (i.e., so a mother who tested positive at 25 weeks gestation, would be matched to a mother who did not test positive for infection during just before or during pregnancy but had an ongoing pregnancy at 25 weeks gestation).

Supplementary Table 1 shows the key sociodemographic and clinical characteristics of the infected pregnancies and the pool of controls. Supplementary Figure 1 shows the number of these pregnancies included in the matched cohorts for each of our infection analyses (i.e., with separate analyses examining each outcome). Supplementary Figure 2 shows the distribution of the infections by calendar time.

Women with confirmed infection between six weeks preconception up to the end of pregnancy were more likely than uninfected controls to be from deprived, and urban, areas (Table 1). Most exposed women (98.9%) had only one confirmed infection in the pregnancy exposure period (Table 2).

After adjusting for covariates, in primary analyses we found an association between maternal infection and preterm birth (adjusted Odds Ratio [aOR] = 1.36, 95% Confidence Interval [CI] = 1.16–1.59). The overall increased risk of preterm birth reflected an increased risk of both spontaneous (aOR = 1.27, 95% CI = 1.03–1.56) and provider-initiated (aOR = 1.42, 95% CI = 1.11–1.81) preterm birth. We also found an association between infection and very preterm birth (aOR = 1.90, 95% CI = 1.20–3.02) and provider-initiated very preterm birth (aOR = 2.63, 95% CI = 1.23–5.62), but not spontaneous very preterm birth (aOR = 1.31, 95% CI = 0.78–2.18) (Table 3). Further detail on the lag between infection (or matching in controls) and delivery is provided in Supplementary Table 2 and on the sub-type of provider-initiated preterm births (induction or pre-labour caesarean section) in Supplementary Table 3.

Due to small numbers of neonatal deaths, we only accounted for matching factors in the model examining this outcome. The point estimate for the association between infection and neonatal death suggested an increased risk (OR ≥ 2), but we had wide CIs spanning the null value (OR = 2.25, 95% CI = 0.95–5.34). We found no evidence of an association between maternal infection and the risk of stillbirth, extended perinatal death, small for gestational age, very small for gestational age, low Apgar score, or very low Apgar score (Table 3). As shown in Supplementary Table 4, risk ratios calculated using conditional Poisson regression showed negligible difference from the ORs calculated using conditional logistic regression.

In subgroup analyses examining the association between maternal infection at ≥20 + 0 gestation and neonatal outcomes, we again found an association between later infection and any preterm birth, spontaneous preterm birth, provider-initiated preterm birth, any very preterm birth, and provider-initiated very preterm birth. Point estimates for the association between later infection and both stillbirth (aOR=2.01, 95% CI = 0.92–4.39) and neonatal death (OR = 2.57, 95% CI = 0.86–7.65) were raised, however the CIs spanned the null value. We did not find an association between early infection (at <20 + 0 gestation) and increased risk of any adverse neonatal outcomes (Fig. 1 and Supplementary Tables 5, 6).

### SARS-CoV-2 infection and maternal outcomes
In primary analyses, we found evidence for an association between maternal infection and admission to critical care or death (aOR = 1.72, 95% CI = 1.39–2.12) and venous thromboembolism (aOR = 2.53, 95% CI = 1.47–4.35) (Table 4). We found no evidence for an association between infection and an increased risk of hypertensive disorders of pregnancy or pregnancy-related bleeding (Table 4 and Supplementary Table 7). As shown in Supplementary Table 4, there is negligible difference between the risk ratios and the ORs.

In subgroup analyses, we again found an association between later infection (at ≥20 + 0 gestation) and critical care admission or death (aOR = 2.12, 95% CI = 1.67-2.69), and venous thromboembolism (aOR = 2.94, 95% CI = 1.53–5.63). We did not find an association between early infection (at <20 + 0 gestation) and increased risk of any adverse maternal outcomes (Fig. 1 and Supplementary Tables 5, 6).

### COVID-19 vaccination and neonatal and maternal outcomes
The COPS study database contained information on 55,167 singleton pregnancies that reached at least 20 + 0 weeks gestation, ended in a live or stillbirth, and where the pregnancy was ongoing or conceived after the start of the COVID-19 vaccination programme on December 8, 2020, and conceived before June 2, 2021. Of these, 11,379 had vaccination (and no confirmed infection) between six weeks preconception up to the end of pregnancy. 1429 had both vaccination and infection during the pregnancy exposure period and were excluded. An

 

**Table 1 | Key characteristics of exposed and control pregnancies included in infection and vaccination analyses[a]**

|  | Infected | Uninfected controls | Vaccinated | Unvaccinated controls |
|---|---|---|---|---|
| Number of pregnancies | 4074 | 12222 | 11379 | 22758 |
| Median maternal age (min–max) | 29 (14–45) | 29 (14–46) | 32 (14–52) | 31 (15–53) |
| *Maternal deprivation* |  |  |  |  |
| 1 (most deprived) | 1225 (30.1%) | 3089 (25.3%) | 1620 (14.2%) | 4890 (21.5%) |
| 2 | 1028 (25.2%) | 2556 (20.9%) | 1838 (16.2%) | 4449 (19.5%) |
| 3 | 681 (16.7%) | 2295 (18.8%) | 2138 (18.8%) | 4196 (18.4%) |
| 4 | 675 (16.6%) | 2404 (19.7%) | 2864 (25.2%) | 5100 (22.4%) |
| 5 (least deprived) | 465 (11.4%) | 1865 (15.3%) | 2917 (25.6%) | 4104 (18%) |
| Unknown/Missing | 0 (0%) | 13 (0.1%) | 2 (<0.1%) | 19 (0.1%) |
| *Maternal ethnicity* |  |  |  |  |
| White | 3611 (88.6%) | 10108 (82.7%) | 9845 (86.5%) | 18851 (82.8%) |
| South Asian | 183 (4.5%) | 383 (3.1%) | 433 (3.8%) | 825 (3.6%) |
| Black/Caribbean/African | 57 (1.4%) | 205 (1.7%) | 132 (1.2%) | 479 (2.1%) |
| Other/mixed ethnicity | 156 (3.8%) | 437 (3.6%) | 468 (4.1%) | 909 (4.0%) |
| Unknown/Missing | 67 (1.6%) | 1089 (8.9%) | 501 (4.4%) | 1694 (7.4%) |
| *Maternal urban/rural status* |  |  |  |  |
| Large urban areas | 1616 (39.7%) | 4136 (33.8%) | 4369 (38.4%) | 8219 (36.1%) |
| Other urban areas | 1644 (40.4%) | 4664 (38.2%) | 3548 (31.2%) | 8167 (35.9%) |
| Accessible small towns | 248 (6.1%) | 980 (8.0%) | 961 (8.4%) | 1842 (8.1%) |
| Remote small towns | 89 (2.2%) | 402 (3.3%) | 346 (3.0%) | 656 (2.9%) |
| Accessible rural areas | 373 (9.2%) | 1375 (11.3%) | 1459 (12.8%) | 2632 (11.6%) |
| Remote rural areas | 81 (2.0%) | 582 (4.8%) | 601 (5.3%) | 1073 (4.7%) |
| Unknown/Missing | 23 (0.6%) | 83 (0.7%) | 95 (0.8%) | 169 (0.7%) |
| *Parity* |  |  |  |  |
| 0 | 1655 (40.6%) | 5430 (44.4%) | 4824 (42.4%) | 8528 (37.5%) |
| 1+ | 2298 (56.4%) | 6435 (52.7%) | 6164 (54.2%) | 13512 (59.4%) |
| Unknown/missing | 121 (3.0%) | 357 (2.9%) | 391 (3.4%) | 718 (3.2%) |
| *Maternal clinical vulnerability* |  |  |  |  |
| Not clinically vulnerable | 3012 (73.9%) | 8698 (73.4%) | 8411 (73.9%) | 16680 (73.3%) |
| Clinically vulnerable | 1035 (25.4%) | 3141 (25.7%) | 2839 (24.9%) | 5879 (25.8%) |
| Extremely vulnerable | 27 (0.7%) | 113 (0.9%) | 129 (1.1%) | 199 (0.9%) |
| *Maternal diabetes* |  |  |  |  |
| No diabetes | 3749 (92.0%) | 11430 (93.5%) | 10411 (91.5%) | 20864 (91.7%) |
| Pre-existing diabetes | 28 (0.7%) | 108 (0.9%) | 147 (1.3%) | 182 (0.8%) |
| Gestational diabetes | 297 (7.3%) | 684 (5.6%) | 821 (7.2%) | 1712 (7.5%) |
| *Maternal smoking status* |  |  |  |  |
| Non-smoker | 2785 (68.4%) | 7976 (65.3%) | 8612 (75.7%) | 15548 (68.3%) |
| Ex-smoker | 819 (20.1%) | 2505 (20.5%) | 2074 (18.2%) | 4509 (19.8%) |
| Smoker | 468 (11.5%) | 1687 (13.8%) | 679 (6.0%) | 2638 (11.6%) |
| Unknown/Missing | 2 (< 0.1%) | 54 (0.4%) | 14 (0.1%) | 63 (0.3%) |
| *Body mass index* |  |  |  |  |
| Underweight | 90 (2.2%) | 337 (2.8%) | 176 (1.5%) | 451 (2.0%) |
| Healthy weight | 1411 (34.6%) | 4701 (38.5%) | 4324 (38%) | 8422 (37.0%) |
| Overweight | 1288 (31.6%) | 3551 (29.1%) | 3431 (30.2%) | 7044 (31.0%) |
| Obese/severely obese | 1203 (29.5%) | 3304 (27.0%) | 3183 (28%) | 6298 (27.7%) |
| Unknown/Missing | 82 (2.0%) | 329 (2.7%) | 265 (2.3%) | 543 (2.4%) |

[a]The data presented related to Cohort 1: singleton pregnancies ending in a live or stillbirth at ≥20 + 0 gestation used for analyses of stillbirth and extended perinatal death outcomes. The exposed pregnancies included in Cohorts 2–10 differed slightly from those in Cohort 1, and the controls were redrawn for each cohort. Characteristics of Cohorts 2–10 therefore differ from those shown above, although differences were minimal.

additional 4005 had infection during the pregnancy exposure period, leaving a pool of 38,354 unvaccinated (and uninfected) control pregnancies for matching to the vaccinated pregnancies on maternal age at conception and gestational week at exposure/matching.

Supplementary Table 1 shows the key sociodemographic and clinical characteristics of the vaccinated pregnancies and the pool of controls. Supplementary Figure 3 shows the number of these pregnancies included in the matched cohorts for each of our vaccination analyses. Supplementary Figure 4 shows the distribution of the vaccinations by calendar time and dose number.

Women receiving vaccination between six weeks preconception up to the end of pregnancy were less likely than unvaccinated controls

**Table 2 | Exposure characteristics in infected and vaccinated cohorts[a]**

| | Infected | Vaccinated |
|---|---|---|
| Number of pregnancies | 4074 | 11379 |
| *Gestation at first exposure* | | |
| Preconception | 220 (5.4%) | 1620 (14.2%) |
| 2–19 weeks | 1080 (26.5%) | 3960 (34.8%) |
| ≥20 weeks | 2774 (68.1%) | 5799 (51.0%) |
| *Number of infections during exposure period* | | |
| 1 | 4030 (98.9%) | – |
| 2 | 44 (1.1%) | – |
| *Timing of infections (among neonates exposed to two infections in pregnancy)* | | |
| <20weeks | 3 (6.8%) | – |
| ≥20 weeks | 10 (22.7%) | – |
| Both | 31 (70.5%) | – |
| *Number of vaccinations during exposure period* | | |
| 1 | – | 3867 (34.0%) |
| 2+ | – | 7512 (66.0%) |
| *Vaccine type* | | |
| Oxford-AstraZeneca ChAdOx1-s/nCoV-19 | – | 785 (6.9%) |
| Moderna mRNA-1273 | – | 1217 (10.7%) |
| Pfizer-BioNTech BNT162b2 | – | 8672 (76.2%) |
| Mixed Doses | – | 705 (6.2%) |

[a]The data presented related to Cohort 1: singleton pregnancies ending in a live or stillbirth at ≥20 + 0 gestation used for analyses of stillbirth and extended perinatal death outcomes. The exposed pregnancies included in Cohorts 2–10 differed slightly from those in Cohort 1. Characteristics of Cohorts 2–10 therefore differ from those shown above, although differences were minimal.

to be from deprived areas and less likely to be smokers (Table 1). Most women received only Pfizer-BioNTech BNT162b2 vaccine (76.2%; Table 2). After adjusting for covariates, we found no evidence that maternal COVID-19 vaccination was associated with increased risk of any of the neonatal or maternal outcomes (Tables 5, 6 and Supplementary Table 8), and there was negligible difference between the risk ratios and the ORs (Supplementary Table 9).

## Discussion

In this national, population-based, matched cohort study we found that SARS-CoV-2 infection just before or during pregnancy was associated with an increased risk of preterm birth (driven by increased risk of both spontaneous and provider-initiated preterm birth), very preterm birth (driven by provider-initiated very preterm birth), maternal critical care admission or death, and maternal venous thromboembolism. Our point estimate for the association between infection and neonatal death was raised, but wide confidence intervals spanned the null hence we could not rule out no association. We found no evidence of an association between infection and increased risk of the other outcomes examined: stillbirth, extended perinatal mortality, small-for-gestational age, low Apgar score, hypertensive disorders of pregnancy and pregnancy-related bleeding. We did not find any evidence that COVID-19 vaccination just before or during pregnancy was associated with an increased risk of any of the neonatal or maternal outcomes examined.

In subgroup analyses, we found an association between infection in later pregnancy (at ≥20 + 0 gestation) and an increased risk of preterm birth (any, spontaneous, and provider-initiated), very preterm birth (any and provider-initiated), maternal critical care admission or death, and maternal venous thromboembolism. We found raised point estimates for the association between later infection and both stillbirth and neonatal death, however again the confidence intervals spanned

the null value. We found no evidence for an association between infection in earlier pregnancy (at <20 + 0 gestation) and an increased risk of any of the adverse neonatal or maternal outcomes examined.

Our study adds to the existing evidence on SARS-CoV-2 infection during pregnancy and neonatal and maternal outcomes. Our results align with those of a living systematic review and meta-analysis that found, based on literature identified up to 27 April 2021, that pregnant women with SARS-CoV-2 infection had increased odds of maternal death (OR = 6.09, 95% CI = 1.82–20.38), admission to the intensive care unit (OR = 5.41, 95% CI = 3.59–8.14), preterm birth (OR = 1.57, 95% CI = 1.36–1.81), stillbirth (OR = 1.81, 95% CI = 1.38–2.37), and admission to the neonatal intensive care unit (OR = 2.18, 95% CI = 1.46–3.26)[2]. Our findings align with current recommendations that pregnant women with SARS-CoV-2 infection should undergo risk assessment for venous thromboembolism[17]. A previous systematic review reported an increased risk of hypertensive disorders following SARS-CoV-2 infection in pregnancy[1], and a US study reported an increased risk of foetal growth restriction and postpartum haemorrhage[18]; however, we do not find any evidence of increased risk of hypertensive disorders of pregnancy, small for gestational age, or pregnancy-related bleeding following infection. The systematic review included 21 studies, the majority of which were non-population-based observational (generally cohort) studies using data from single or multiple healthcare providers to compare pregnancy-related outcomes of women with, compared to without, SARS-CoV-2 infection in pregnancy. How infection and outcomes were defined and ascertained is not specified. Three of 16 studies including preeclampsia as an outcome reported a significant association between infection and preeclampsia. Infection was also found to be significantly associated with the risk of preeclampsia on meta-analysis of results from all 16 studies. The US study used data from a single insurance provider to conduct a retrospective cohort study of enroled women with, compared to without SARS-CoV-2 infection in pregnancy. Infection status was mainly ascertained through clinical diagnosis rather than viral testing results, and adjustment for confounders was limited (no adjustment for parity or maternal smoking or body mass index [BMI]).

There is limited existing evidence on the nature of any increased risk of preterm birth following infection, and on how the timing of infection during pregnancy influences subsequent outcomes. We have shown that the increased risk of preterm birth (i.e., at <37 + 0 gestation) following infection reflects an increased risk of both spontaneous and provider-initiated preterm birth, whereas the increased risk of very preterm birth (i.e., at <32 + 0 gestation) reflects an increased risk of provider-initiated preterm birth only. A US surveillance study found that SARS-CoV-2 infection during the third trimester was associated with a higher frequency of preterm birth compared to infections during the first and second trimesters[19]. In subgroup analyses, we have shown that the increased risk of adverse neonatal and maternal outcomes is restricted to infections occurring in later (at ≥20 + 0 gestation), compared to earlier (at <20 + 0 gestation) pregnancy. On balance therefore, our findings do not currently indicate a need for ongoing enhanced maternal or foetal surveillance during pregnancy or delivery following SARS-CoV-2 infection early in pregnancy, however further research on this is warranted. Other evidence, including from the COPS study, shows no increased risk of early pregnancy loss[7] or congenital anomalies[8] following maternal infection in early pregnancy. Evidence is currently lacking on any long-term impacts on children's health and development following in-utero infection.

Our findings add to the growing body of evidence on the safety of COVID-19 vaccinations during pregnancy. A population-based retrospective cohort study in Ontario Canada including more than 43,000 births to individuals vaccinated during pregnancy found that COVID-19 vaccination during pregnancy was not associated with any increased risk of overall preterm birth, spontaneous preterm birth, very preterm

**Table 3 | Association between exposure to SARS-CoV-2 during pregnancy exposure period and neonatal outcomes, calculated using conditional logistic regression**

| Cohort (Outcome) | Infection status | Number of pregnancies/ neonates | Number with outcome | % with outcome | Odds ratio accounting for only matching[a] [95% CI] | P value[a] | Adjusted odds ratio[b] [95% CI] | P value[b] |
|---|---|---|---|---|---|---|---|---|
| 1 (Stillbirths) | Infected | 4074 | 14 | 0.3% | 1.02 [0.55–1.88] | 0.94 | 1.08 [0.56–2.05] | 0.82 |
|  | Uninfected | 12,222 | 41 | 0.3% | Ref | – | Ref | - |
| 1 (Extended perinatal death) | Infected | 4074 | 23 | 0.5% | 1.13 [0.70–1.82] | 0.62 | 1.13 [0.68–1.89] | 0.63 |
|  | Uninfected | 12,222 | 61 | 0.5% | Ref | – | Ref | - |
| 2 (Neonatal death) | Infected | 4060 | 9 | 0.2% | 2.25 [0.95–5.34] | 0.07 | – | - |
|  | Uninfected | 12,180 | 12 | 0.1% | Ref | – | – | – |
| 3 (Small for gestational age (<10th percentile)[c]) | Infected | 3947 | 185 | 4.5% | 0.91 [0.77-1.08] | 0.26 | 1.04 [0.86-1.26] | 0.67 |
|  | Uninfected | 11,532 | 592 | 5.0% | Ref | – | Ref | – |
| 3 (Very small for gestational age (<3rd percentile)[c]) | Infected | 3947 | 38 | 0.9% | 0.88 [0.61-1.26] | 0.48 | 1.02 [0.65-1.61] | 0.93 |
|  | Uninfected | 11,532 | 126 | 1.1% | Ref | – | Ref | – |
| 4 (Low Apgar score (<7)[c]) | Infected | 3663 | 64 | 1.7% | 0.99 [0.74-1.32] | 0.95 | 0.96 [0.71-1.30] | 0.80 |
|  | Uninfected | 10,594 | 184 | 1.6% | Ref | – | Ref | – |
| 4 (Very low Apgar score (<4)[c]) | Infected | 3663 | 10 | 0.3% | 0.77 [0.38–1.54] | 0.46 | 0.87 [0.42–1.81] | 0.72 |
|  | Uninfected | 10,594 | 37 | 0.3% | Ref | – | Ref | – |
| 5 (Preterm birth) | Infected | 3603 | 274 | 7.6% | 1.37 [1.18-1.59] | <0.001 | 1.36 [1.16-1.59] | <0.001 |
|  | Uninfected | 10,809 | 611 | 5.7% | Ref | – | Ref | – |
| 5 (Spontaneous preterm birth[c]) | Infected | 3597 | 149 | 4.1% | 1.25 [1.03-1.52] | 0.03 | 1.27 [1.03-1.56] | 0.03 |
|  | Uninfected | 10,787 | 361 | 3.3% | Ref | - | Ref | - |
| 5 (Provider-initiated preterm birth[c]) | Infected | 3597 | 119 | 3.3% | 1.46 [1.17-1.82] | <0.001 | 1.42 [1.11-1.81] | 0.005 |
|  | Uninfected | 10,787 | 246 | 2.3% | Ref | – | Ref | – |
| 6 (Very preterm birth) | Infected | 2842 | 38 | 1.3% | 1.68 [1.13–2.49] | 0.01 | 1.90 [1.20–3.02] | 0.01 |
|  | Uninfected | 8526 | 68 | 0.8% | Ref | – | Ref | – |
| 6 (Spontaneous very preterm birth[c]) | Infected | 2841 | 22 | 0.8% | 1.29 [0.77–2.13] | 0.31 | 1.31 [0.78–2.18] | 0.31 |
|  | Uninfected | 8522 | 51 | 0.6% | Ref | – | Ref | – |
| 6 (Provider-initiated very preterm birth[c]) | Infected | 2841 | 15 | 0.5% | 2.81 [1.39–5.69] | <0.001 | 2.63 [1.23–5.62] | 0.01 |
|  | Uninfected | 8522 | 16 | 0.2% | Ref | – | Ref | – |

*CI* confidence interval.

[a]Matched for maternal age, gestation at infection/matching and seasons of conception.

[b]Adjustment for deprivation only for analyses with the following outcomes: spontaneous very preterm birth and provider-initiated very preterm birth. Adjustment for parity and deprivation only for analyses with the following outcomes: stillbirth, extended perinatal death, and very low Apgar score. Adjustment for all covariates apart from parity for the following outcomes: preterm birth, spontaneous preterm birth, provider-initiated preterm birth and very preterm birth. Adjustment for all covariates for analyses for the following outcomes: small for gestational age, very small for gestational age, and low Apgar score.

[c]For these outcomes there is not an exact match of three uninfected controls to each neonate exposed to infection as neonates with missing outcome data have been removed from analysis; see Methods for further details on missing data.

birth, small for gestational age at birth, or stillbirth[20]. Similarly, a systematic review has reported that there was no evidence of a higher risk of maternal outcomes including hypertensive disorders of pregnancy, pulmonary embolism, postpartum haemorrhage, maternal death, or intensive care unit admission among those receiving COVID-19 vaccination during pregnancy[21].

The strengths of this study include the use of a national linked dataset with comprehensive data available on many key socio-demographic and clinical covariates. We have been able to include a wide range of neonatal and maternal outcomes and, through a standardised methodological approach, have been able to provide evidence on any risk associated with SARS-CoV-2 infection (in the absence of vaccination) and, separately, COVID-19 vaccination (in the absence of infection) which is essential to allow balanced consideration of risks and benefits.

Our study has some important limitations. We relied on the results of viral testing to identify confirmed SARS-CoV-2 infections. Testing policy in Scotland evolved throughout the pandemic[22,23]. For our infection analyses, we restricted our study period to the date from which widespread community testing for symptomatic individuals was implemented (May 18, 2020), to minimise the impact of the substantial under-ascertainment of symptomatic infections associated with restricted testing prior to that point. It is still possible however that we may have under-ascertained some (generally mild) symptomatic infections if individuals did not get tested. In addition to testing of individuals with symptoms, routine testing of asymptomatic individuals was made increasingly available as the pandemic progressed, for example routine testing of all individuals admitted to hospital was implemented from December 2020. It is likely therefore that the confirmed infections included in our study will include both

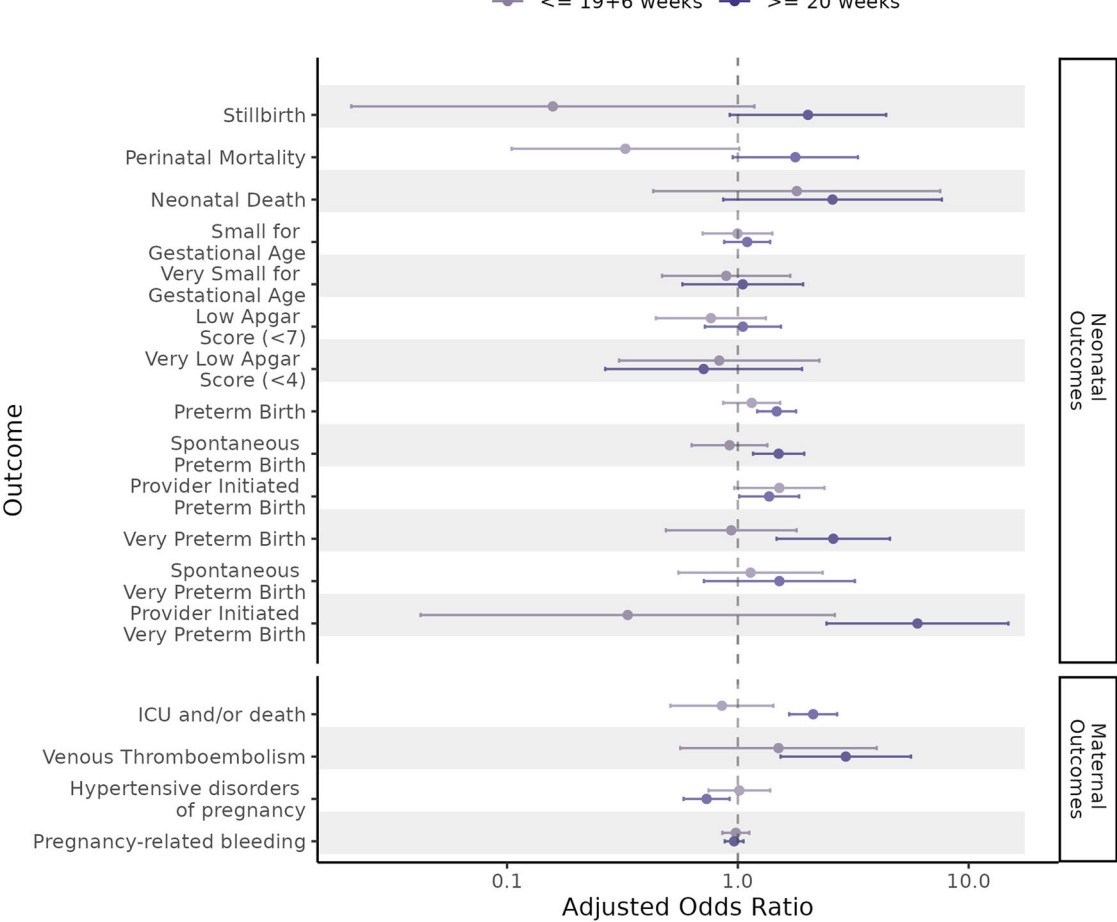

**Fig. 1 | Adjusted odds ratios for the association between SARS-CoV-2 infection and neonatal and maternal outcomes, stratified by timing of first SARS-CoV-2 infection in the pregnancy exposure period.** The sample size for the analysis for each outcome is provided in Supplementary Table 5 and Supplementary Table 6. Odds ratios (represented by the points on the plot) were calculated using conditional logistic regression. The errors bars for each data point are 95% confidence intervals.

symptomatic and asymptomatic infections. We did not have reliable data on symptoms or severity of infection however, so we could not provide information on the relative numbers or explore how association with adverse outcomes varied by symptom status. Also, we did not have access to negative test results within the COPS study database, hence could not provide information on overall testing rates. As noted in the methods, there was a change in testing guidelines in early January 2022 meaning confirmatory PCR testing following a positive lateral flow device (LFD) test was no longer necessary to confirm infection, but this is unlikely to have affected our results given that it only covered a very short period towards the end of our study and LFD tests were still widely available. We did not have reliable data on viral variant so we could not explore whether the associations varied by variant. The infections in pregnancy that are included in our analyses occurred over a relatively wide time period, from May 2020 to February 2022. It is therefore likely that included infections were caused by a range of viral variants, from wild type through Alpha, Delta, and Omicron as these were sequentially dominant in Scotland across this time frame. We have previously shown that infections in the period when Omicron (B.1.1.529) was the dominant variant were associated with a lower risk of adverse neonatal and maternal outcomes compared with infections when Delta (B.1.617.2) was the dominant variant[24]. Despite using national data, there were small absolute numbers of some outcomes (such as neonatal death), precluding fully adjusted analyses and leading to high levels of uncertainty in our estimates of association. However, we report all results in line with our

study protocol as pooling of results across different studies will give more precise estimates. Regarding the adjustment of covariates, our analysis was not able to include some potentially important confounders such as exposure to environmental tobacco smoke due to the lack of recording within electronic health records. We were also unable to match pregnancies on exact calendar time (i.e., matching exposed and control neonates by the calendar week of estimated conception) due to an insufficient pool of controls, but were able to match on season of conception for our infection analyses. Most of the existing evidence on the safety of COVID-19 vaccines in pregnancy, including that provided by our study, relates to mRNA vaccines. Further research on the safety of other vaccine types, including viral vector vaccines, would be beneficial. Lastly, our study does not assess the effectiveness of COVID-19 vaccination in preventing, or moderating, pregnancy-related outcomes following SARS-CoV-2 infection in pregnancy. Understanding the effectiveness of vaccines in pregnancy is an important area of research, and there is a growing body of evidence to suggest that vaccination may reduce the risk of some adverse neonatal and maternal outcomes by reducing the risk among women who are infected with SARS-CoV-2 infection[6,24].

In summary, our national, population-based study provides high-quality evidence that SARS-CoV-2 infection in pregnancy is associated with increased risk of serious neonatal and maternal adverse outcomes, including preterm birth, very preterm birth, maternal critical care admission or death, and maternal venous thromboembolism. We expand on previous studies to show that the increased risk of preterm

**Table 4 | Association between exposure to SARS-CoV-2 during pregnancy exposure period and maternal outcomes, calculated using conditional logistic regression**

| Cohort (Outcome) | Infection status | Number of pregnancies/neonates | Number with outcome | % with outcome | Odds ratio accounting for only matching[a] [95% CI] | P value[a] | Adjusted odds ratio[b] [95% CI] | P value[b] |
|---|---|---|---|---|---|---|---|---|
| 7 (Critical care and/or Death) | Infected | 4074 | 175 | 4.3% | 1.83 [1.51-2.21] | <0.001 | 1.72 [1.39-2.12] | <0.001 |
| | Uninfected | 12222 | 290 | 2.4% | Ref | – | Ref | – |
| 8 (Venous thromboembolism) | Infected | 4072 | 29 | 0.7% | 2.72 [1.65-4.49] | <0.001 | 2.53 [1.47-4.35] | <0.001 |
| | Uninfected | 12216 | 32 | 0.3% | Ref | – | Ref | – |
| 9 (Hypertensive disorders of pregnancy) | Infected | 4071 | 178 | 4.4% | 0.85 [0.72-1.01] | 0.06 | 0.82 [0.68-0.98] | 0.03 |
| | Uninfected | 12213 | 624 | 5.1% | Ref | – | Ref | – |
| 10 (Pregnancy-related bleeding) | Infected | 3973 | 1665 | 41.9% | 0.96 [0.90-1.05] | 0.33 | 0.97 [0.90-1.04] | 0.39 |
| | Uninfected | 11919 | 5101 | 42.8% | Ref | –– | Ref | – |

CI confidence interval

[a]Matched for maternal age, gestation at infection/matching and seasons of conception and additionally for health board of residence (Greater Glasgow & Clyde or Lanarkshire versus all other health boards) for hypertensive disorders of pregnancy and pregnancy-related bleeding.

[b]Adjustment for parity and deprivation only for analyses with the following outcomes: venous thromboembolism. Adjustment for all covariates for analyses for the following outcomes: maternal critical care or death, hypertensive disorders of pregnancy and pregnancy-related bleeding.

birth reflects the increased risk of both spontaneous and provider-initiated preterm birth, whereas the increased risk of very preterm birth reflects the increased risk of provider-initiated very preterm birth only. We also show that the increased risks are associated with infections in later, but not earlier, pregnancy. We do not find any evidence of an increased risk of hypertensive disorders of pregnancy or pregnancy-related bleeding following infection at any stage of pregnancy. Together, these findings can help to inform the clinical care of women with SARS-CoV-2 infection in pregnancy, for example, they suggest a need for vigilance for venous thromboembolism but no clear need for enhanced ongoing pregnancy monitoring due to infection in early pregnancy. We found no evidence that COVID-19 vaccination just before or during pregnancy was associated with any adverse neonatal or maternal outcomes, supporting current recommendations that COVID-19 vaccination remains the safest and most effective way for pregnant women to protect themselves and their babies from the risks associated with SARS-CoV-2 infection.

## Methods

### Study design

We conducted a population-based, matched cohort study following a study protocol and statistical analysis plan which can be accessed on GitHub (https://github.com/Public-Health-Scotland/COPS-public[25]). Reporting followed the Strengthening the Reporting of Observational studies in Epidemiology (STROBE) checklist[26].

### Setting and participants

This study used the mid-July 2022 update of the COPS database[15,16]. This includes all recognised pregnancies in Scotland from January 1, 2015 onwards, identified through linkage of the following national records: antenatal care booking; General Practitioner (GP) records; general acute hospital discharges (Scottish Morbidity Record (SMR) 01); maternity hospital discharges (SMR 02); statutory termination of pregnancy notifications (AAS); National Records of Scotland (NRS) statutory live and stillbirth registrations; and NHS live birth notifications. For each pregnancy, comprehensive data is available, including on the estimated date of conception, gestational age at the end of the pregnancy, the pregnancy outcome (miscarriage, ectopic pregnancy, molar pregnancy, termination of pregnancy, stillbirth, live birth, unknown pregnancy outcome, and ongoing pregnancy), and maternal clinical and socio-demographic characteristics.

For this study, we included singleton pregnancies that reached at least 20 weeks, 0 days (20 + 0) gestation and ended before 44 weeks, 6 days (44 + 6) gestation in a live or stillbirth. We use the term "neonate" as the overarching term including these live and stillbirths. For our infection analyses, we included pregnancies ongoing or conceived after the start of widespread community testing for SARS-CoV-2 on May 18, 2020. For our vaccination analyses, we included pregnancies ongoing or conceived after the start of the COVID-19 vaccination programme on December 8, 2020. Live born neonates were followed up to four weeks (28 days) after birth and all mothers up to six weeks (42 days) postpartum. To allow sufficient time for the return and incorporation of records relating to the end of pregnancy plus the four-week neonatal or six-week postpartum period prior to data extraction, we excluded pregnancies conceived after June 1, 2021, to ensure all included pregnancies could be followed-up to the likely upper gestational limit of 40 + 6 gestation by February 28, 2022. However, to minimise unnecessary loss of data, pregnancies conceived towards the end of the inclusion period were retained even if they delivered after 40 + 6 (up to 44 + 6) and hence a small number of pregnancies ended in March 2022.

### Outcomes

We examined the following neonatal outcomes: stillbirth, neonatal death, extended perinatal mortality, low Apgar score (5 min Apgar

**Table 5 | Association between exposure to COVID-19 vaccination during pregnancy exposure period and neonatal outcomes, calculated using conditional logistic regression**

| Cohort (outcome) | Vaccination Status | Number of pregnancies/ neonates | Number with outcome | % with outcome | Odds ratio accounting for only matching[a] [95% CI] | P value[a] | Adjusted odds ratio[b] [95% CI] | P value[b] |
|---|---|---|---|---|---|---|---|---|
| 1 (Stillbirths) | Vaccinated | 11379 | 29 | 0.3% | 1.08 [0.68–1.70] | 0.75 | 1.03 [0.63–1.68] | 0.91 |
| | Unvaccinated | 22758 | 54 | 0.2% | Ref | – | Ref | – |
| 1 (Extended perinatal death) | Vaccinated | 11379 | 41 | 0.4% | 0.92 [0.63–1.34] | 0.66 | 0.97 [0.64-1.48] | 0.90 |
| | Unvaccinated | 22758 | 89 | 0.4% | Ref | – | Ref | – |
| 2 (Neonatal death) | Vaccinated | 11350 | 12 | 0.1% | 0.56 [0.29–1.06] | 0.07 | 0.51 [0.26-1.01] | 0.05 |
| | Unvaccinated | 22700 | 43 | 0.2% | Ref | – | Ref | – |
| 3 (Small for gestational age (<10th percentile)[c]) | Vaccinated | 10951 | 455 | 4.2% | 0.89 [0.79–0.99] | 0.04 | 1.02 [0.90-1.16] | 0.79 |
| | Unvaccinated | 21267 | 987 | 4.6% | Ref | – | Ref | – |
| 3 (Very small for gestational age (<3rd percentile)[c]) | Vaccinated | 10951 | 103 | 0.9% | 0.90 [0.71–1.14] | 0.40 | 1.06 [0.81-1.40] | 0.66 |
| | Unvaccinated | 21267 | 223 | 1.0% | Ref | – | Ref | – |
| 4 (Low Apgar score (<7)[c]) | Vaccinated | 10286 | 176 | 1.7% | 1.08 [0.90-1.30] | 0.41 | 1.13 [0.93-1.37] | 0.23 |
| | Unvaccinated | 19761 | 313 | 1.6% | Ref | – | Ref | – |
| 4 (Very low Apgar score (<4)[c]) | Vaccinated | 10286 | 26 | 0.3% | 1.11 [0.68–1.80] | 0.67 | 1.32 [0.79–2.23] | 0.29 |
| | Unvaccinated | 19761 | 44 | 0.2% | Ref | – | Ref | – |
| 5 (Preterm birth) | Vaccinated | 11202 | 598 | 5.3% | 0.93 [0.84-1.03] | 0.15 | 0.96 [0.86-1.06] | 0.41 |
| | Unvaccinated | 22404 | 1283 | 5.7% | Ref | – | Ref | – |
| 5 (Spontaneous preterm birth[c]) | Vaccinated | 11193 | 356 | 3.1% | 0.89 [0.79-1.01] | 0.08 | 0.95 [0.83-1.08] | 0.43 |
| | Unvaccinated | 22362 | 794 | 3.6% | Ref | – | Ref | – |
| 5 (Provider-initiated preterm birth[c]) | Vaccinated | 11193 | 233 | 2.1% | 1.00 [0.86-1.18] | 0.97 | 0.95 [0.79-1.13] | 0.55 |
| | Unvaccinated | 22362 | 464 | 2.1% | Ref | – | Ref | – |
| 6 (Very preterm birth) | Vaccinated | 10057 | 74 | 0.7% | 0.88 [0.67-1.16] | 0.36 | 0.89 [0.64-1.23] | 0.47 |
| | Unvaccinated | 20114 | 168 | 0.8% | Ref | – | Ref | – |
| 6 (Spontaneous very preterm birth[c]) | Vaccinated | 10052 | 56 | 0.6% | 0.93 [0.68-1.28] | 0.67 | 0.92 [0.63-1.35] | 0.67 |
| | Unvaccinated | 20101 | 120 | 0.6% | Ref | – | Ref | – |
| 6 (Provider-initiated very preterm birth[c]) | Vaccinated | 10052 | 13 | 0.1% | 0.58 [0.31-1.07] | 0.08 | 0.69 [0.35-1.28] | 0.22 |
| | Unvaccinated | 20101 | 45 | 0.2% | Ref | – | Ref | – |

*CI* confidence interval

[a]Matched for maternal age and gestation at vaccination/matching.

[b]Adjustment for deprivation only for analyses with the following outcomes: provider-initiated very preterm birth. Adjustment for parity and deprivation only for analyses with the following outcomes: stillbirth, neonatal death, and very low Apgar score. Adjustment for all covariates apart from parity for the following outcomes: preterm birth, spontaneous preterm birth, provider-initiated preterm birth, very preterm birth and spontaneous very preterm birth. Adjustment for all covariates for analyses for the following outcomes: extended perinatal death, small for gestational age, very small for gestational age, and low Apgar score.

[c]For these outcomes there is not an exact match of two unvaccinated controls to each neonate exposed to vaccination as neonates with missing outcome data have been removed from analysis; see Methods for further details of missing data.

score <7), very low Apgar score (5 min Apgar score <4), small-for-gestational age (birthweight <10th centile), very small-for-gestational age (birthweight <3rd centile), preterm birth (<37 + 0 weeks gestation) and very preterm birth (<32 + 0 weeks gestation). For preterm births and very preterm births, we examined these overall and by the type of onset of preterm birth (spontaneous or provider-initiated). NRS neonatal death records were linked to the COPS database using unique identifiers to ascertain neonatal deaths. Stillbirths and other neonatal outcomes were available in the COPS database, derived from NRS stillbirth and SMR02 delivery records. We examined the following maternal outcomes: admission to critical (intensive or high dependency) care (any cause) or death (any cause), hypertensive disorders of pregnancy, pregnancy-related bleeding, and venous thromboembolism. General acute (SMR01), maternity (SMR02), and critical care (Scottish Intensive Care Society Audit Group, SICSAG) discharge

records and NRS death records were linked to the COPS database to ascertain these outcomes occurring during pregnancy or the post-partum period. Further detail on our outcome measures is provided in Table 7 and Supplementary Table 10.

**Exposures**

National data on confirmed SARS-CoV-2 infections and COVID-19 vaccinations were linked to the COPS database to identify exposures of interest. For infection analyses, we identified confirmed infections by a positive SARS-CoV-2 reverse transcription polymerase chain reaction (RT-PCR) test or, from Jan 6, 2022, a positive LFD test (unless the LFD result was followed by a negative RT-PCR result within 48 h)[27]. Tests taken in hospital, in community testing centres, and (for LFD tests) at home, in response to symptoms or as part of regular/routine testing of asymptomatic individuals, were all included. A subsequent positive

**Table 6 | Association between exposure to COVID-19 vaccination during pregnancy exposure period and maternal outcomes, calculated using conditional logistic regression**

| Cohort (outcome) | Vaccination Status | Number of pregnancies/ neonates | Number with outcome | % with outcome | Odds ratio accounting for only matching[a] [95% CI] | P value[a] | Adjusted odds ratio[b] [95% CI] | P value[b] |
|---|---|---|---|---|---|---|---|---|
| 7 (Critical care and/or Death) | Vaccinated | 11379 | 253 | 2.2% | 0.92 [0.79–1.07] | 0.26 | 0.86 [0.73–1.01] | 0.07 |
| | Unvaccinated | 22758 | 551 | 2.4% | Ref | – | Ref | – |
| 8 (Venous thromboembolism) | Vaccinated | 11375 | 17 | 0.1% | 0.60 [0.35–1.04] | 0.07 | 0.62 [0.35–1.09] | 0.10 |
| | Unvaccinated | 22750 | 56 | 0.2% | Ref | – | Ref | – |
| 9 (Hypertensive disorders of pregnancy) | Vaccinated | 11373 | 638 | 5.6% | 0.99 [0.90–1.10] | 0.89 | 0.89 [0.80–0.99] | 0.04 |
| | Unvaccinated | 22746 | 1284 | 5.6% | Ref | – | Ref | – |
| 10 (Pregnancy-related bleeding) | Vaccinated | 11230 | 5195 | 46.3% | 1.09 [1.04–1.14] | <0.001 | 1.04 [0.99–1.09] | 0.14 |
| | Unvaccinated | 22460 | 9900 | 44.1% | Ref | – | Ref | – |

CI confidence interval

[a]Matched for maternal age and gestation at vaccination/matching and additionally for health board of residence (Greater Glasgow & Clyde or Lanarkshire versus all other health boards) for hypertensive disorders of pregnancy and pregnancy-related bleeding.
[b]Adjustment for parity and deprivation only for analyses with the following outcomes: venous thromboembolism. Adjustment for all covariates for the following outcomes: maternal critical care or death, hypertensive disorders of pregnancy and pregnancy-related bleeding.

test during the relevant exposure period was considered a separate infection if it was >90 days after a prior infection. For vaccination analyses, we identified vaccinations with any vaccine type (Oxford/ AstraZeneca ChAdOx1-S/nCoV-19, Pfizer-BioNTech BNT162b2, or Moderna mRNA-1273) or dose number (first to fourth). All vaccinations given during the relevant exposure period were included. Pregnancies were considered exposed if they had a confirmed infection, or vaccination, between six weeks preconception up to the end of the relevant outcome-specific exposure period: up to 31 + 6 gestation for very preterm birth, 36 + 6 gestation for preterm birth, or the end of pregnancy for all other outcomes (Table 7).

We did not include exposures more than six weeks prior to conception when classifying exposure status, as such prior exposures were uncommon. For example, only 3.1% (127 of 4,074) of the women with infection during the pregnancy exposure period included in our analyses had had infection, vaccination, or both more than six weeks prior to conception.

### Covariates

Data were available on the following covariates: maternal area-level deprivation, rural urban status, ethnicity, clinical vulnerability, diabetes, smoking status, BMI and parity. Maternal area-level deprivation was based on maternal postcode of residence mapped to Scottish Index of Multiple Deprivation (SIMD) quintile. SIMD ranks small areas across Scotland on the basis of administrative data relating to income, employment, education, health, access to services, crime, and housing[28,29]. Areas are then categorised into quintiles from 1 (including the most deprived fifth of the population) to 5 (the least deprived fifth). Maternal rural urban status was based on maternal postcode mapped to the Scottish urban-rural categorisation. The urban-rural categorisation classifies settlements across Scotland based on population size and (for rural and remote areas) drive time to the nearest urban area[30]. A detailed categorisation was used for descriptive analyses (large urban areas, other urban areas, accessible small towns, remote small towns, accessible rural areas, remote rural areas, and unknown). A less detailed categorisation (urban, rural, unknown) was used for adjustment in models. Ethnicity was based on self-reported ethnicity included on healthcare records and grouped in five categories according to the Scottish decennial population census categories (White, South Asian, Black/Caribbean/African, other/mixed ethnicity, and unknown)[31].

Women were grouped as clinically extremely vulnerable, clinically vulnerable, or not clinically vulnerable. Women were classified as clinically extremely vulnerable if they were identified on the national highest risk/shielding list maintained by Public Health Scotland[32] and, of those not extremely vulnerable, were classified as clinically vulnerable if they were in any Q-COVID risk group[33] (excluding diabetes) or had hypertension according to cross-sectional GP/primary care data available from June 2020 and January 2021. To categorise women by diabetes status (pre-existing diabetes, gestational diabetes or unknown onset, no diabetes, or unknown), data were taken from SMR02 maternity discharge records where possible; if this was not available then data were extracted from GP records. For smoking status (smoker, ex-smoker, non-smoker, and unknown) and BMI (underweight<18.5 kg/m$^2$, healthy weight 18.5-< 25, overweight 25-< 30 and obese/severely obese≥30), data were taken from SMR02 delivery records where possible (which provide information on smoking status and BMI at antenatal booking); if this was not available then these data were extracted from GP records. There was one exception to this for smoking—if it was documented that a woman was a non-smoker at antenatal booking, but they were recorded as either a smoker or ex-smoker on a GP record, then we categorised the woman as an ex-smoker. For parity, data were taken from SMR02 maternity discharge records, with parity calculated as the total number of previous pregnancies minus the total number of spontaneous and therapeutic abortions.

**Table 7 | Definition, relevant exposure period, and additional eligibility criteria for each outcome measure**

| Cohort | Outcome | Outcome definition | Outcome-specific exposure period (for infection or vaccination) | Additional eligibility criteria[a] |
|---|---|---|---|---|
| 1 | Stillbirth | Spontaneous foetal loss at ≥20 + 0 gestation from any cause | From 6 weeks preconception to end of pregnancy | No additional criteria |
| 1 | Extended perinatal mortality | Stillbirth or neonatal death from any cause | From 6 weeks preconception to end of pregnancy | No additional criteria |
| 2 | Neonatal mortality | Death of live born neonate within 28 days of birth from any cause | From 6 weeks preconception to end of pregnancy | Live births only |
| 3 | Small-for-gestational age | Birthweight <10th centile by WHO-UK90 growth reference | From 6 weeks preconception to end of pregnancy | Live births from 23 + 0 to 42 + 6 gestation only |
| 3 | Very small-for-gestational age | Birthweight <3rd centile by WHO-UK90 growth reference | From 6 weeks preconception to end of pregnancy | Live births from 23 + 0 to 42 + 6 gestation only |
| 4 | Low Apgar score | 5 min Apgar score <7 | From 6 weeks preconception to end of pregnancy | Live births at ≥37 + 0 gestation only |
| 4 | Very low Apgar score | 5 min Apgar score <4 | From 6 weeks preconception to end of pregnancy | Live births at ≥37 + 0 gestation only |
| 5 | Preterm birth | Births <37 + 0 gestation | From 6 weeks preconception to 36 + 6 gestation | Live births only |
| 5 | Spontaneous preterm birth | Births <37 + 0 gestation following P-PROM or spontaneous onset of labour reported on delivery records | From 6 weeks preconception to 36 + 6 gestation | Live births only |
| 5 | Provider-initiated preterm birth | Births <37 + 0 gestation following induction of labour or pre-labour caesarean section reported on delivery records or, in the absence of these data, where delivery occurred during a critical care admission | From 6 weeks preconception to 36 + 6 gestation | Live births only |
| 6 | Very preterm birth | Births <32 + 0 gestation | From 6 weeks preconception to 31 + 6 weeks gestation | Live births only |
| 6 | Spontaneous very preterm birth | Births <32 + 0 gestation following P-PROM or spontaneous onset of labour reported on delivery records | From 6 weeks preconception to 31 + 6 weeks gestation | Live births only |
| 6 | Provider-initiated very pre-term birth | Births <32 + 0 gestation following induction of labour or pre-labour caesarean section reported on delivery records or, in the absence of these data, where delivery occurred during a critical care admission | From 6 weeks preconception to 31 + 6 weeks gestation | Live births only |
| 7 | Maternal critical care admission and/or death | Critical (intensive or high dependency) care admission as identified on SICSAG record or death from any cause following exposure/matching during pregnancy and within 42 days of the end of pregnancy as identified in NRS death records | From 6 weeks preconception to end of pregnancy | No additional criteria |
| 8 | Maternal venous thromboembolism | Venous thromboembolism recorded on general acute, maternity, or critical care discharge record during pregnancy or within 42 days of the end of pregnancy[b] | From 6 weeks preconception to end of pregnancy | Excluding any pregnancies with venous thromboembolism before exposure |
| 9 | Maternal hypertensive disorders of pregnancy | Hypertensive disorder of pregnancy recorded on general acute, maternity, or critical care discharge record during pregnancy or within 42 days of the end of pregnancy[b] | From 6 weeks preconception to end of pregnancy | Excluding any pregnancies with hypertensive disorders of pregnancy before exposure |
| 10 | Maternal pregnancy-related bleeding | Pregnancy-related bleeding recorded on general acute, maternity, or critical care discharge record during pregnancy or within 42 days of the end of pregnancy[b] | From 6 weeks preconception to end of pregnancy | Excluding any pregnancies with pregnancy-related bleeding before exposure |

[a]Beyond core study inclusion criteria (singleton pregnancies ending in live or stillbirth at ≥20 + 0 gestation from relevant date range).
[b]See Supplementary Table 10 for the diagnostic codes indicating these outcomes.
P-PROM Preterm Premature Rupture Of Membranes; SICSAG Scottish Intensive Care Society Audit Group.

## Creating matched cohorts

Separate cohorts, and matched controls, were drawn for each outcome measure (for infection and vaccination analyses separately) due to the differing eligibility criteria (Table 7).

To create matched cohorts for our infection analyses, we matched each pregnancy with confirmed infection (and no vaccination) in the relevant exposure period to three pregnancies with no confirmed infection (or vaccination) in the exposure period. Pregnancies where the woman had both infection and vaccination in the relevant exposure period were excluded from analysis. Pregnancies were matched on maternal age at conception (+/- one year), season of conception, and gestational week of first infection in the exposed pregnancy (hence a pregnancy where the mother was infected at 25 weeks gestation would be matched to a pregnancy reaching at least 25 weeks gestation). Controls were selected randomly for each exposed pregnancy, starting with the exposed pregnancies with the smallest number of potential controls and iterating through each exposed pregnancy with the pool of controls getting increasingly smaller. In a deviation from our protocol, we additionally matched on health board of residence (Greater Glasgow & Clyde and Lanarkshire versus all other health boards) in cohorts examining two maternal outcomes: hypertensive disorders of pregnancy and pregnancy-related bleeding. This was due to consistently lower recording of these outcomes on maternity (SMR02) discharge records in Greater Glasgow & Clyde and Lanarkshire, potentially reflecting regional differences in clinical pathways or recording practices (Supplementary Figs. 5, 6).

Some of our outcomes had missing data (small for gestational age, type of onset of preterm birth [provider-initiated or spontaneous preterm birth] and Apgar score). While the neonates with missing data are likely to be different to neonates without missing data (for example, potentially the sickest neonates), the proportion of neonates missing data for these outcomes was low and was similar in our exposed and control cohorts (see Supplementary Table 11). We therefore dropped exposed neonates with missing outcome data along with their matched controls, and dropped any additional controls with missing outcome data, for the cohorts examining these outcomes.

Creation of matched cohorts for vaccination analyses was similar. We matched each pregnancy with vaccination (and no infection) in the relevant exposure period to two pregnancies with no vaccination (or infection) in the exposure period. Pregnancies where the woman had both infection and vaccination in the relevant exposure period were again excluded from analysis. Pregnancies were matched on maternal age at conception and gestational week of first vaccination. Only two matched controls were selected for each vaccinated pregnancy (compared to three for each infected pregnancy), due to the smaller pool of unvaccinated controls, largely reflecting the shorter study period for the vaccination analysis compared with the infection analysis.

## Statistical analysis

We conducted descriptive analyses examining covariates, exposures, and outcomes in exposed and control cohorts. In line with our prespecified protocol, we assessed associations between our exposures and outcomes using conditional logistic regression. We first ran crude models with no covariates (thereby only accounting for the matching factors). For models with sufficient outcome events (≥30), we then ran models adjusting for covariates (all covariates if ≥90 outcome events, deprivation and parity only if 30–89 outcome events)[34]. In general, for covariates with missing data, we included "unknown" categories in our models (see Table 1). We did this to prevent unbalanced matching as a result of dropping some exposed or control pregnancies due to missing data on covariates. We did, however, conduct complete case analyses dropping any pregnancies with missing covariate information to reassure that this was not biasing our results and found negligible changes to our results. For models examining preterm birth, we did not adjust for parity due to relatively high levels of missing parity data

among preterm births within our infected cohort. For models examining hypertensive disorders of pregnancy and pregnancy-related bleeding, where cohorts were additionally matched on health board, we confirmed that the association between exposures and these outcomes was similar in Glasgow/Lanarkshire and all other health boards before presenting an overall combined adjusted estimate. In response to reviewer request, in a deviation from our protocol, we replicated these analyses using conditional Poisson regression allowing us to calculate risk ratios for comparison with the odds ratios from the conditional logistic regression models. Analyses were conducted in R 3.6.1., with code developed for this analysis available on GitHub[25].

## Subgroup analyses

We conducted stratified analyses of the association between infection and each outcome by timing of the first infection during the relevant exposure period (< or ≥ 20 + 0 gestation). Pregnancies where the woman had infections at < and ≥ 20 + 0 weeks gestation were excluded.

## Ethics and permissions

COPS has ethical approval from the National Research Ethics Service Committee, South East Scotland 02 (REC 12/SS/0201: SA 2) and information governance approval from the Public Benefit and Privacy Panel for Health and Social Care (2021-0116).

## Reporting summary

Further information on research design is available in the Nature Portfolio Reporting Summary linked to this article.

# Data availability

Aggregate data files on COVID-19 vaccinations and SARS-CoV-2 infections among pregnant women are available here: https://www.opendata.nhs.scot/organization/health_protection. Patient-level data underlying this article cannot be shared publicly due to data protection and confidentiality requirements. Public Health Scotland is the data holder for the data used in this study. Data can be made available to approved researchers for analysis after securing relevant permissions from the data holders via the Public Benefit and Privacy Panel. Enquiries regarding data availability should be directed to phs.edris@phs.scot.

# Code availability

Metadata and code are available on GitHub at https://github.com/Public-Health-Scotland/COPS-public[35].

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

## Acknowledgements

Our thanks to the EAVE II Patient Advisory Group and Sands charity for their support. COPS is a sub-study of EAVE II, which is funded by the Medical Research Council (MC_PC_19075) with the support of BREATHE—The Health Data Research Hub for Respiratory Health [MC_PC_19004], which is funded through the UK Research and Innovation Industrial Strategy Challenge Fund and delivered through Health Data Research UK. Additional support has been provided through Public Health Scotland and Scottish Government DG Health and Social Care and the Data and Connectivity National Core Study, led by Health Data Research UK in partnership with the Office for National Statistics and funded by UK Research and Innovation. COPS has received additional funding from Tommy's charity. SJS is funded by a Wellcome Trust Clinical Career Development Fellowship (209560/Z/17/Z). SVK acknowledges funding from a NRS Senior Clinical Fellowship (SCAF/15/02), the Medical Research Council (MC_UU_00022/2) and the Scottish Government Chief Scientist Office (SPHSU17).

## Author contributions

R.W., S.J.S., C.R. and A.S. conceived the study. R.W., S.J.S., C.C., C.R., and A.S. designed the study. R.W., S.J.S. and C.C. drafted the protocol. J.C., C.D., J.D., L.E.M.H., L.H., A.G., L.L., T.M., E.M. and B.T. prepared the dataset for analysis. L.L. and C.C. performed data analysis. T.S. conducted the literature review. L.L., C.C., R.W., S.J.S., T.S. and A.S. prepared the first draft of the manuscript. L.L., C.C., T.S., J.C., C.D., J.D., L.E.M.H., L.H., A.G., T.M., E.M., B.T., K.B., S.V.K., R.M., C.M., C.S., C.R., A.S., R.W. and S.J.S. provided critical input to drafts of the manuscript. R.W., S.J.S., C.C. and A.S. gave final approval for the version to be published. RW acts as a guarantor for the study.

## Competing interests

A.S. and C.R. were members of the Scottish Government's COVID-19 Advisory Group. A.S. is a member of the New and Emerging

Respiratory Virus Threats Advisory Group (NERVTAG) risk stratification subgroup and the Scottish Government's Committee on Pandemic Preparedness. C.R. is a member of the Scientific Pandemic Influenza Group on Modelling and the MHRA Covid 19 Vaccine Benefit and Risk Working Group. AS is a member of AstraZeneca's Thrombotic Thrombocytopenic Advisory Group. All roles are unremunerated. S.V.K. was co-chair of Scottish Government's Expert Reference Group on Ethnicity and COVID-19. All other authors declare no competing interests.

## Additional information

Laura Lindsay [1], Clara Calvert [2,3], Ting Shi [2], Jade Carruthers[1], Cheryl Denny [1], Jack Donaghy[1], Lisa E. M. Hopcroft [1,4], Leanne Hopkins [1], Anna Goulding [1,5], Terry McLaughlin[1], Emily Moore [1], Bob Taylor [1], Krishnan Bhaskaran [3], Srinivasa Vittal Katikireddi [1,6], Ronan McCabe[6], Colin McCowan [7], Colin R. Simpson [2,8], Chris Robertson [1,9], Aziz Sheikh[2], Rachael Wood [1,2,10] ✉ & Sarah J. Stock [1,2,10]

[1]Public Health Scotland, Edinburgh, Scotland, UK. [2]Usher Institute, University of Edinburgh, Edinburgh, UK. [3]Faculty of Epidemiology and Population Health, London School of Hygiene and Tropical Medicine, London, UK. [4]National Services Scotland, Edinburgh, Scotland, UK. [5]Gloucestershire Hospitals NHS Foundation Trust, Gloucester, UK. [6]MRC/CSO Social & Public Health Sciences Unit, University of Glasgow, Glasgow, UK. [7]School of Medicine, University of St Andrews, St Andrews, UK. [8]Wellington Faculty of Health, Victoria University of Wellington, Wellington, New Zealand. [9]Department of Mathematics and Statistics, University of Strathclyde, Glasgow, UK. [10]These authors jointly supervised this work: Rachael Wood, Sarah J. Stock. ✉e-mail: Rachael.Wood@phs.scot

