## [Peer Review File · Nature Communications]

Neonatal and maternal outcomes following SARS-CoV-2 infection and COVID-19 vaccination: a population-based matched cohort studyREVIEWER COMMENTS

Reviewer #1 (Remarks to the Author):

This population-based study examined the association between infections and vaccination occurring from 6 weeks prior to pregnancy onset date through end of pregnancy in relation to pregnancy and maternal outcomes. For each exposure of interest, exposed women were matched to unexposed women.

SARS-CoV-2 infection 6 months prior to conception or during pregnancy was associated with preterm birth, maternal admission to critical care or death, and venous thromboembolism. The study also found a non-statistically elevated risk of neonatal death associated with SARS-CoV-2 infection. However, there were no association between maternal vaccination and increased risk of preterm birth. There were some suggestions of a protective effect of vaccination against SGA. This is a large study with a statistical analysis appropriate for the design.

Here are my few comments for consideration:

1. What was the rationale for including infections that occurred 6 weeks prior to conception in the evaluation of maternal and pregnancy outcome? I am not sure that an infection occurring 6 weeks prior to conception would affect a pregnancy outcome.
2. For maternal outcomes like hospital admission, it would be important to examine the length of time between the infection and when this outcome occur to make sure that the hospitalization was related to infection. Do the investigators verify that hospitalization was due to COVID-19?
3. What was the percentage of infection occurring before and during pregnancy?
4. I strongly suggest a sensitivity analysis restricting infections to those occurring from pregnancy onset and see if the findings still hold.
5. Women who were vaccinated and infected were excluded. However, it would have been of interest to determine outcomes among this group especially when the infection occurred after vaccination.
6. Covariates: It would be helpful if the investigators can provide more details on clinical vulnerability and the other covariates like smoking. How were these collected in the database? Self-reported? What is the unit for BMI?
7. Line 185: what do you mean by onset of preterm?
8. You conducted a subgroup analysis looking at infection <20 week and =>20 weeks. What was the reason for not conducting the same subgroup analysis for vaccination?
9. I am worried about the protective effect of the vaccine for SGA. I am wondering if this is due to immortal time bias.
10. Lines 345- 349: The investigators should point out the difference between their study and those of previous studies with different results.
11. Line 353 – 355: The investigators make a difference between spontaneous preterm birth and provider-initiated preterm birth. What is the clinical difference between these two in relation to infection? Did the clinicians initiated the preterm birth to save the mom and baby or was this planned in advance regardless of infection? Was clinician initiated preterm birth done by C-section or vaginal?
12. Line 260 – 262: I think the claim of no need for enhance surveillance early in pregnancy is very premature. It is possible that early infections could lead to developmental abnormalities.
13. Table 4: there is a typo in the title. "shading" instead of "sharing"

Reviewer #2 (Remarks to the Author):

This is a very important manuscript outlining potential maternal, pregnancy, and infant outcomes associated with SARS-CoV-2 infection and COVID-19 vaccination immediately before or during pregnancy. This manuscript further strengthens recommendations for COVID-19 vaccination during pregnancy and highlights the risks to mothers and infants associated with infection during pregnancy. The representative nature of data collection throughout Scotland and the cohort design of the study are major strengths. I have the following comments:

To enhance clarity, throughout the paper, the authors could consider using the term "infant" or

"neonate" instead of "baby," as infancy is typically defined as those <12 months of age.

Lines 79-80: Seems like an oddly worded sentence. Consider replacing "To help" with "Helping"

Line 107: Gestational age at the end of pregnancy? Can you also specify here pregnancy outcomes under study in parentheses? (Are these health outcomes for the mother, or do they include items like stillbirth and live births?) While outcomes are described in detail later and thoroughly in Table 1, it would be helpful to have a description when they are first introduced.

Line 26: Why is "very" in parentheses? Can you define what these values are for the reader? I.e., what is considered very low Apgar score? Very small for gestational age? Etc... These are nicely defined in Table 1, but could be helpful to define in the text of the methods section.

Lines 145-146: Did you collect dates of vaccination, and consider the impact of timing of vaccination during pregnancy on pregnancy and infant outcomes?

Line 159-160: Rural or urban status? (Or was an urban/rural index of some sort used?) Was BMI determined pre-pregnancy? How was clinical vulnerability determined?

Lines 170-173: Were pregnancies matched more stringently on a certain date, e.g. LMP or date the pregnancy ended? Important since there are implications for expansion of both infection-induced seroprevalence and vaccination coverage during the exposure period.

Line 205: Very important. It is unclear why conditional logistic regression was used for primary analyses when this is a matched cohort study. Other matched modeling methods, such as conditional Poisson, are available and would provide the appropriate measure (risk ratios) for a matched cohort study instead of odds ratios, which may overestimate the effect.

In the results and discussion sections:
Very nice Table 1. Helpful to the reader.

Table 3: Consider noting reference groups explicitly where model estimates are provided (e.g., "Ref" instead of "1")

Figure 1: Could the authors consider using alternatively white/gray shaded rows in their forest plot to make the figure a little bit easier to read? (This can be done fairly quickly in the forester package in R.)

Line 209: Did the authors consider imputing missing data, and comparing complete-case to imputed results to assess bias?

Line 225: Please specify "weeks" when discussing before or after 20 weeks of gestation.

Line 298: Suspect incorrect date. Did you intend to state that your COVID-19 vaccination program began on December 8, 2021?

Lines 341-343: As Allotey et al. are continuously updating their findings, you could probably withhold providing the OR and 95%CI estimates and simply state the outcomes that infection during pregnancy are associated with.

Line 421: Suggest modifying to state "no clear need for enhanced ongoing pregnancy monitoring due to infection in early pregnancy." (Essentially, to imply that antenatal care is still vitally important.)

Reviewer #3 (Remarks to the Author):

Review of manuscript – NCOMMS-23-16089

Thank you for the opportunity to review this manuscript. This is a national study from Scotland on the impact of SARS-CoV-2 infection and COVID-19 vaccination on infant and maternal outcomes. Infection was associated with preterm birth, maternal admission to critical care or death and VTE. There was no increased risk associated with vaccination against COVID-19. The study is of high quality using national Scottish data.

General comments

1. It is not clear what proportion of women were tested for SARS-CoV-2 infection and when during pregnancy this was performed. From previous studies we know that many women who tested positive at delivery did not have symptoms of COVID-19. Were only women with symptoms tested? Were all women tested on admission to delivery? Was there information also on negative tests?
2. Was it possible to address the various variants of SARS-CoV-2 viruses during the study period. Research has found different results based on the variants. It is stated that the study matched for season of conception but did this also include the variants of the virus?
3. It is likely that women with signs of preterm birth were more often tested and this could influence the findings. Hence, in parallel with comment no 1, testing strategy is likely to affect the results.

Specific comments

1. It is unclear why pregnancies conceived towards the end of the inclusion period were retained. Could this not lead to a selection of pregnancies and hence influence the results?
2. The adjusted OR for hypertensive disease of pregnancy was 0.82 (0.68-0.98). Still it is stated that infection was not associated with hypertensive disease of pregnancy? Please revise and comment on this finding.
3. In the conclusion it is stated that increased risks of adverse outcomes are associated with infection in later, not earlier. This has been shown in previous studies and would hence not be stated as novel.

REVIEWER COMMENTS

Reviewer #1 (Remarks to the Author):

This population-based study examined the association between infections and vaccination occurring from 6 weeks prior to pregnancy onset date through end of pregnancy in relation to pregnancy and maternal outcomes. For each exposure of interest, exposed women were matched to unexposed women.

SARS-CoV-2 infection 6 months prior to conception or during pregnancy was associated with preterm birth, maternal admission to critical care or death, and venous thromboembolism. The study also found a non-statistically elevated risk of neonatal death associated with SARS-CoV-2 infection. However, there were no association between maternal vaccination and increased risk of preterm birth. There were some suggestions of a protective effect of vaccination against SGA. This is a large study with a statistical analysis appropriate for the design.

Here are my few comments for consideration:

1. What was the rationale for including infections that occurred 6 weeks prior to conception in the evaluation of maternal and pregnancy outcome? I am not sure that an infection occurring 6 weeks prior to conception would affect a pregnancy outcome.

Author response: At the start of the COPS study, to inform development of our project protocol we reviewed methodological guidance on vaccine safety analyses, including from the World Health Organization and the Global Alignment of Immunization Safety Assessment in Pregnancy. WHO guidance on assessment of COVID-19 vaccine safety in pregnancy available at that time specifically recommended applying an exposure window starting a month before the last menstrual period i.e. 6 weeks prior to conception. Unfortunately, this guidance has subsequently been withdrawn hence we cannot provide a link to it. Guidelines from GAIA (for example Pathways to preterm birth: Case definition and guidelines for data collection, analysis, and presentation of immunization safety data - ScienceDirect & Hypertensive disorders of pregnancy: Case definitions & guidelines for data collection, analysis, and presentation of immunization safety data - ScienceDirect) more generally suggest that the exposure period for vaccine safety analyses should extend from just before/around conception to the end of pregnancy and/or that arbitrary exposure periods should not be chosen. In general this guidance reflects the fact that immunological changes around the time of conception may plausibly influence subsequent placental development/function and hence later pregnancy outcomes such as preterm birth. This guidance is therefore relevant for analyses examining outcomes following infection as well as vaccination.

2. For maternal outcomes like hospital admission, it would be important to examine the length of time between the infection and when this outcome occur to make sure that the hospitalization was related to infection. Do the investigators verify that hospitalization was due to COVID-19?

Author response: As noted in Table 1, our outcome measure examining maternal critical care admission (and/or death) included any admission to intensive or high dependency care, for any cause, at any point from the point of exposure/matching during pregnancy to within 42 days of the end of pregnancy. In the infected cohort, this outcome would therefore include any admissions directly due to maternal COVID-19 disease, any that are due to related (e.g. thromboembolism) or unrelated (e.g. bleeding) pregnancy-related complications, and any due to other incidental conditions. Similarly, in the vaccinated cohort, this outcome would include any admissions due to adverse effects of vaccination, should any occur. In the control cohorts, this outcome will capture the 'background rate' of critical care admissions due to pregnancy-related complications and incidental conditions. Our results therefore provide an estimate of the overall risk of critical care admission (or death) following infection or vaccination, relative to the background rate. The excess risk found following infection cannot be assumed to be solely due to the immediate effects of

maternal COVID-19 disease, for example some may be due to the increased risk of specific pregnancy-related complications (in particular thromboembolism) associated with maternal infection.

3. What was the percentage of infection occurring before and during pregnancy?

Author response: As reported in Table 2, in our infected cohort, 5.4% (220/4,074) of first infections occurring during the pregnancy exposure period occurred in the six weeks before conception, with the remainder (94.6%, 3,854/4,074) of first infections occurring from conception to the end of pregnancy.

4. I strongly suggest a sensitivity analysis restricting infections to those occurring from pregnancy onset and see if the findings still hold.

Author response: We did consider including a sensitivity analysis restricting to only infections occurring from pregnancy onset but for the following reasons do not think this is necessary in the paper:

1. As reported in the response to the previous comment, only a small percentage of our infection exposures occurred in the 6 weeks preconception so any sensitivity analysis excluding this group is unlikely to show any substantial changes to our main results.
2. We already include subgroup analyses by timing of infection in pregnancy, in which we show that infection in later pregnancy (from 20 weeks gestation onwards) is associated with adverse outcomes but infection in 'early pregnancy' (from 6 weeks preconception to 19 weeks gestation inclusive) is not. Even in our 'early pregnancy' exposure group, only a minority of infections (209/1,269, 16.5%) only occurred in the preconception period (11 women had infection in both the preconception period and later in pregnancy), hence an additional analysis excluding this group is unlikely to substantially change these results.

5. Women who were vaccinated and infected were excluded. However, it would have been of interest to determine outcomes among this group especially when the infection occurred after vaccination.

Author response: This is an interesting issue that we discussed and considered at length in the development of our protocol. As our main objective was to provide information on the balance of risk of having SARS-CoV-2 infection in pregnancy and the safety of COVID-19 vaccination in pregnancy we decided to conduct clean analyses looking at the impacts of these different in-pregnancy exposures separately. We expressly did not attempt to examine how vaccination may moderate outcomes following infection as this is examining vaccine effectiveness rather than vaccine safety and was outwith the scope of our already extensive study. A key reason for not doing this analysis with the COPS data is we would have a relatively small sample size (for such an analysis, our "exposed group" of women who had infection following vaccination in pregnancy would be 1080) and therefore we would not have sufficient power to look at many of our outcomes of interest. We have noted that we do not attempt to look at vaccine effectiveness in the study in the Introduction, as follows:

"Helping pregnant women and their healthcare providers make informed decisions on the importance of COVID-19 vaccination in pregnancy requires high-quality data on: (1) the impact of SARS-CoV-2 infection (in the absence of vaccination) on adverse maternal and baby outcomes; (2) the safety of vaccination in pregnancy; and (3) the effectiveness of vaccination in reducing the impact of the infection (either by preventing infection or reducing the severity of infection). In our study, we examined the first two of these three issues."

We also note that it was beyond the scope of our study to look at vaccine effectiveness but this is an important area for further work in the Discussion:

“Lastly, our study does not assess the effectiveness of COVID-19 vaccination in preventing, or moderating pregnancy-related outcomes following, SARS-CoV-2 infection in pregnancy. Understanding the effectiveness of vaccines in pregnancy is an important area of research, and there is a growing body of evidence to suggest that vaccination may reduce the risk of some adverse baby and maternal outcomes by reducing the risk among women who are infected with SARS-CoV-2 infection^{6,34}.”

6. Covariates: It would be helpful if the investigators can provide more details on clinical vulnerability and the other covariates like smoking. How were there collected in the database? Self-reported? What is the unit for BMI?

Author response: We had previously included information on covariates in a supplementary table, but it is clear based on this comment and a similar comment from Reviewer 2 that it would be helpful to include this in the main manuscript. We have therefore substantially expanded the description of the covariates in the Methods section and removed Supplementary Table 2. The text in the Covariates section in the Methods now reads (additions underlined):

“Data were available on the following covariates: maternal area-level deprivation, rural urban status, ethnicity, clinical vulnerability, diabetes, smoking status, body mass index (BMI) and parity. Maternal area-level deprivation was based on maternal postcode of residence mapped to Scottish Index of Multiple Deprivation (SIMD) quintile. SIMD ranks small areas across Scotland on the basis of administrative data relating to income, employment, education, health, access to services, crime, and housing^{20,21}. Areas are then categorised into quintiles from 1 (including the most deprived fifth of the population) to 5 (the least deprived fifth). Maternal rural urban status was based on maternal postcode mapped to the Scottish urban-rural categorisation. The urban-rural categorisation classifies settlements across Scotland based on population size and (for rural and remote areas) drive time to the nearest urban area²². A detailed categorisation was used for descriptive analyses (large urban areas, other urban areas, accessible small towns, remote small towns, accessible rural areas, remote rural areas, and unknown). A less detailed categorisation (urban, rural, unknown) was used for adjustment in models. Ethnicity was based on self-reported ethnicity included on healthcare records and grouped in five categories according to the Scottish decennial population census categories (White, South Asian, Black/Caribbean/African, other/mixed ethnicity and unknown)²³.

Women were grouped as clinically extremely vulnerable, clinically vulnerable, or not clinically vulnerable. Women were classified as clinically extremely vulnerable if they were identified on the national highest risk/shielding list maintained by Public Health Scotland²⁴ and, of those not extremely vulnerable, were classified as clinically vulnerable if they were in any Q-COVID risk group²⁵ (excluding diabetes) or had hypertension according to cross-sectional GP/primary care data available from June 2020 and January 2021. To categorise women by diabetes status (pre-existing diabetes, gestational diabetes or unknown onset, no diabetes, or unknown), data were taken from SMR02 maternity discharge records where possible; if this was not available then data were extracted from GP records. For smoking status (smoker, ex-smoker, non-smoker, and unknown) and BMI (underweight<18.5kg/m², healthy weight 18.5-<25, overweight 25-<30 and obese/severely obese≥30), data were taken from SMR02 delivery records where possible (which provide information on smoking status and BMI at antenatal booking); if this was not available then these data were extracted from GP records. There was one exception to this for smoking – if it was documented that a woman was a non-smoker at antenatal booking, but they were recorded as either a smoker or ex-smoker on a GP record, then we categorized the woman as an ex-smoker. For parity, data were taken from SMR02 maternity discharge records, with parity calculated as the total number of previous pregnancies minus the total number of spontaneous and therapeutic abortions.”

7. Line 185: what do you mean by onset of preterm?

Author response: We have now provided a clearer description of what we mean by onset of preterm births in the Outcomes section in the Methods (additions underlined):

For preterm births and very preterm births, we examined these overall and by the type of onset of preterm birth (spontaneous or provider-initiated).

We have also revised the sentence highlighted by the reviewer to use the same term “type of onset of preterm birth” so it now reads as follows:

“Some of our outcomes had missing data (small for gestational age, type of onset of preterm birth [provider-initiated or spontaneous preterm birth] and Apgar score).”

Full definitions for spontaneous and provider-initiated preterm births are provided in Table 1. Spontaneous births are those following preterm premature rupture of the membranes or spontaneous onset of labour. Provider-initiated births are those following induction of labour or pre-labour caesarean section reported on delivery records or, in the absence of these data, where delivery occurred during a critical care admission.

8. You conducted a subgroup analysis looking at infection <20 week and =>20 weeks. What was the reason for not conducting the same subgroup analysis for vaccination?

Author response: Before conducting the subgroup analyses, we specified that subgroup analyses by timing of exposure would be undertaken if our primary analyses found any association between exposure (infection or vaccination) at any point in pregnancy and the outcomes studied. We did not find any such associations for vaccination. In addition, in practice we found that the pattern of infection and vaccination exposures during pregnancy were very different, and subgroup analyses examining outcomes following vaccination in early and later pregnancy would not have been methodologically appropriate. Specifically, the proportion of women in our vaccinated cohort who received more than one vaccination during pregnancy (7,512/11,379, 66.0%) was much higher than the proportion of women in our infected cohort who had more than one infection during pregnancy (44/4,074, 1.1%). If we were to do subgroup analyses by vaccination pre and post 20 weeks gestation taking a similar approach to the infection subgroup analyses, we would need to remove any pregnancies where there was exposure to vaccination both pre and post 20 weeks gestation. This would involve exclusion of a high proportion of women receiving multiple vaccinations during pregnancy. As pregnancies where only one vaccination was given are likely to be different to pregnancies where multiple vaccinations were given, this approach would be subject to selection bias.

9. I am worried about the protective effect of the vaccine for SGA. I am wondering if this is due to immortal time bias.

Author response: Our primary analyses find no evidence for an association between vaccination just before or during pregnancy and either small for gestational age (adjusted OR=1.02, 95% CI=0.90-1.16) or very small for gestational age (adjusted OR=1.06, 95% CI=0.81-1.40) after adjusting for key covariates. However, had we found a protective effect, this would not be driven by immortal time bias as we designed our study to eliminate the risk of this. Specifically, we ensured that our exposed babies and their unexposed controls were matched for gestational age at exposure (vaccination or matching) and that all babies included in the study could be followed-up for the outcome. This means that our exposed group and control group were followed up in the same way and there were no time periods between exposure (i.e., vaccination in the exposed group or not being vaccinated but reaching same gestational week as the vaccinated baby they were matched to in the control group) and outcome that were excluded from the study.

10. Lines 345- 349: The investigators should point out the difference between their study and those of previous studies with different results.

Author response: We have amended this section of the Discussion as follows (additions underlined): *“Some previous studies have A previous systematic review reported an increased risk of hypertensive disorders following SARS-CoV-2 infection in pregnancy¹, and a US study reported an increased risk of fetal growth restriction and postpartum haemorrhage²⁸, however we do not find any evidence of increased risk of hypertensive disorders of pregnancy, (very) small for gestational age, or pregnancy-related bleeding following infection. The systematic review included 21 studies, the majority of which were non-population based observational (generally cohort) studies using data from single or multiple healthcare providers to compare pregnancy-related outcomes of women with, compared to without, SARS-CoV-2 infection in pregnancy. How infection and outcomes were defined and ascertained is not specified. Three of 16 studies including preeclampsia as an outcome reported a significant association between infection and preeclampsia. Infection was also found to be significantly associated with risk of preeclampsia on meta-analysis of results from all 16 studies. The US study used data from a single insurance provider to conduct a retrospective cohort study of enrolled women with, compared to without SARS-CoV-2 infection in pregnancy. Infection status was mainly ascertained through clinical diagnosis rather than viral testing results, and adjustment for confounders was limited (no adjustment for parity or maternal smoking or BMI).”*

11. Line 353 - 355: The investigators make a difference between spontaneous preterm birth and provider-initiated preterm birth. What is the clinical difference between these two in relation to infection? Did the clinicians initiated the preterm birth to save the mom and baby or was this planed in advance regardless of infection? Was clinician initiated preterm birth done by C-section or vaginal?

Author response: There are different clinical pathways through which SARS-CoV-2 infection during pregnancy may lead to preterm birth. Inflammatory processes associated with infection may trigger spontaneous preterm birth (ref: <https://www.science.org/doi/10.1126/science.1251816>). Alternatively, infection may necessitate provider-initiated preterm birth to facilitate maternal treatment or resuscitation, or (once fetal viability has been reached) due to concerns about fetal well-being, or a combination of both. Our results show that there is an increased risk of both spontaneous and provider-initiated preterm birth following SARS-CoV-2 infection during pregnancy, suggesting that both key pathways are important. This has implications for prevention and treatment of infection during pregnancy, for example emphasising the need for prevention of infection through vaccination and avoidance of exposure, and prompt and effective treatment of women with infection with interventions that reduce the severity of infection and associated symptoms including fever.

Similar to our response to the reviewer’s second point above, as noted in Table 1, our preterm birth outcomes (all, spontaneous, and provider-initiated) included all preterm births occurring at any point following infection/vaccination or matching up to the relevant gestational cut off. We did not attempt to classify preterm births in our exposed cohorts as directly due to maternal infection (or vaccination) or not, as this is not feasible given the routine data available to us.

We provide information on the lag between infection (or matching) and end of pregnancy for pregnancies in our infected (and uninfected control) cohorts that ended in preterm and very preterm birth in Supplementary Table 4. We have also added an additional supplementary table (Supplementary Table 5) in the revised manuscript, which provides the breakdown of the sub-types of provider-initiated preterm births (induction or pre-labour caesarean section), as is now signposted in the Results (addition underlined):

“Further detail on the lag between infection (or matching in controls) and delivery is provided in Supplementary Table 4 and on the sub-type of provider-initiated preterm births (induction or pre-labour caesarean section) in Supplementary Table 5.”

12. Line 260 - 262: I think the claim of no need for enhance surveillance early in pregnancy is very premature. It is possible that early infections could lead to developmental abnormalities.

Author response: We are not clear whether the reviewer is thinking here primarily of structural congenital anomalies (i.e. is SARS-CoV-2 infection in early pregnancy teratogenic) or post-natal neurodevelopment of children who were exposed to in-utero infection. We agree that both are potentially relevant. There is accumulating evidence of no increased risk of congenital anomalies following SARS-CoV-2 infection in early pregnancy, including from the COPS study here. Evidence on long term neurodevelopment of children exposed to maternal infection in utero is still inevitably sparse given the time lags involved. We have therefore amended the relevant text as follows (additions underlined):

“On balance therefore, our findings do not currently indicate a need for ongoing enhanced maternal or fetal surveillance during pregnancy or delivery following SARS-CoV-2 infection early in pregnancy, however further research on this is warranted. Other evidence, including from the COPS study, shows no increased risk of early pregnancy loss⁷ or congenital anomalies⁸ following maternal infection in early pregnancy. Evidence is currently lacking on any long-term impacts on children’s health and development following in-utero infection.”

13. Table 4: there is a typo in the title. "shading" instead of "sharing"

Author response: Thank you, we have fixed this typo.

Reviewer #2 (Remarks to the Author):

This is a very important manuscript outlining potential maternal, pregnancy, and infant outcomes associated with SARS-CoV-2 infection and COVID-19 vaccination immediately before or during pregnancy. This manuscript further strengthens recommendations for COVID-19 vaccination during pregnancy and highlights the risks to mothers and infants associated with infection during pregnancy. The representative nature of data collection throughout Scotland and the cohort design of the study are major strengths. I have the following comments:

To enhance clarity, throughout the paper, the authors could consider using the term "infant" or "neonate" instead of "baby," as infancy is typically defined as those <12 months of age.

Author response: In preparing the manuscript, we carefully considered the best terminology to use when referring to "baby" outcomes, including using "neonatal" and "infant" outcomes. However, we settled on "baby" as "neonatal" and "infant" strictly only refer to live born babies (thus excluding stillbirths, an important outcome in our study). In addition, as the reviewer notes, "infant" also includes the post-neonatal period (from 28 days of age up to a baby's first birthday), which we did not include in this study.

Lines 79-80: Seems like an oddly worded sentence. Consider replacing "To help" with "Helping":

Author response: Thank you for this suggestion. We have now revised this sentence to read (edit underlined):

"Helping pregnant women and their healthcare providers make informed decisions on the importance of COVID-19 vaccination in pregnancy requires high-quality data on: (1) the impact of SARS-CoV-2 infection (in the absence of vaccination) on adverse maternal and baby outcomes; (2) the safety of vaccination in pregnancy; and (3) the effectiveness of vaccination in reducing the impact of the infection (either by preventing infection or reducing the severity of infection)."

Line 107: Gestational age at the end of pregnancy? Can you also specify here pregnancy outcomes under study in parentheses? (Are these health outcomes for the mother, or do they include items like stillbirth and live births?) While outcomes are described in detail later and thoroughly in Table 1, it would be helpful to have a description when they are first introduced.

Author response: We have revised the sentence as suggested so it now reads (additions underlined):
"For each pregnancy, comprehensive data is available, including on the estimated date of conception, gestational age at the end of the pregnancy, the pregnancy outcome (miscarriage, ectopic pregnancy, molar pregnancy, termination of pregnancy, stillbirth, live birth, unknown pregnancy outcome, and ongoing pregnancy), and maternal clinical and socio-demographic characteristics."

Line 26: Why is "very" in parentheses? Can you define what these values are for the reader? I.e., what is considered very low Apgar score? Very small for gestational age? Etc... These are nicely defined in Table 1, but could be helpful to define in the text of the methods section.

Author response: We have now expanded the description of the outcomes in the Methods, including listing out each outcome individually (i.e., making it clear that low Apgar score and very low Apgar are two different outcomes) and adding in short definitions for these outcomes in parenthesis. This description now reads as follows (additions underlined):

"We examined the following baby outcomes: stillbirth, neonatal death, extended perinatal mortality, low Apgar score (5 minute Apgar score <7), very low Apgar score (5 minute Apgar score <4), small-for-gestational age (birthweight <10th centile), very small-for-gestational age (birthweight <3rd centile), preterm birth (<37+0 weeks gestation) and very preterm birth (<32+0 weeks gestation). For preterm births and very preterm births, we examined these overall and by the type of onset of preterm birth (spontaneous or provider-initiated)."

Lines 145-146: Did you collect dates of vaccination, and consider the impact of timing of vaccination during pregnancy on pregnancy and infant outcomes?

Author response: We do have dates of vaccination available, and these were used to report descriptive information, for example, the gestation at first exposure to vaccination reported in Table 2. As noted in our response to reviewer 1, point 8, we do not think it is appropriate to consider the impact of the timing of vaccination during pregnancy on maternal and baby outcomes due to the large proportion of our vaccinated cohort that received multiple vaccinations during pregnancy.

Line 159-160: Rural or urban status? (Or was an urban/rural index of some sort used?) Was BMI determined pre-pregnancy? How was clinical vulnerability determined?

Author response: As noted in our response to reviewer 1, point 6, we have now moved the following additional information on covariates from the supplementary material to the Methods section in the main manuscript (additions underlined):

“Data were available on the following covariates: maternal area-level deprivation, rural urban status, ethnicity, clinical vulnerability, diabetes, smoking status, body mass index (BMI) and parity. Maternal area-level deprivation was based on maternal postcode of residence mapped to Scottish Index of Multiple Deprivation (SIMD) quintile. SIMD ranks small areas across Scotland on the basis of administrative data relating to income, employment, education, health, access to services, crime, and housing^{20,21}. Areas are then categorised into quintiles from 1 (including the most deprived fifth of the population) to 5 (the least deprived fifth). Maternal rural urban status was based on maternal postcode mapped to the Scottish urban-rural categorisation. The urban-rural categorisation classifies settlements across Scotland based on population size and (for rural and remote areas) drive time to the nearest urban area²². A detailed categorisation was used for descriptive analyses (large urban areas, other urban areas, accessible small towns, remote small towns, accessible rural areas, remote rural areas, and unknown). A less detailed categorisation (urban, rural, unknown) was used for adjustment in models. Ethnicity was based on self-reported ethnicity included on healthcare records and grouped in five categories according to the Scottish decennial population census categories (White, South Asian, Black/Caribbean/African, other/mixed ethnicity and unknown)²³.

Women were grouped as clinically extremely vulnerable, clinically vulnerable, or not clinically vulnerable. Women were classified as clinically extremely vulnerable if they were identified on the national highest risk/shielding list maintained by Public Health Scotland²⁴ and, of those not extremely vulnerable, were classified as clinically vulnerable if they were in any Q-COVID risk group²⁵ (excluding diabetes) or had hypertension according to cross-sectional GP/primary care data available from June 2020 and January 2021. To categorise women by diabetes status (pre-existing diabetes, gestational diabetes or unknown onset, no diabetes, or unknown), data were taken from SMR02 maternity discharge records where possible; if this was not available then data were extracted from GP records. For smoking status (smoker, ex-smoker, non-smoker, and unknown) and BMI (underweight<18.5kg/m², healthy weight 18.5-<25, overweight 25-<30 and obese/severely obese≥30), data were taken from SMR02 delivery records where possible (which provide information on smoking status and BMI at antenatal booking); if this was not available then these data were extracted from GP records. There was one exception to this for smoking – if it was documented that a woman was a non-smoker at antenatal booking, but they were recorded as either a smoker or ex-smoker on a GP record, then we categorized the woman as an ex-smoker. For parity, data were taken from SMR02 maternity discharge records, with parity calculated as the total number of previous pregnancies minus the total number of spontaneous and therapeutic abortions.”

Lines 170-173: Were pregnancies matched more stringently on a certain date, e.g. LMP or date the pregnancy ended? Important since there are implications for expansion of both infection-induced seroprevalence and vaccination coverage during the exposure period.

Author response: We were only able to match each pregnancy with confirmed infection in the pregnancy exposure period to three pregnancies with no confirmed infection on maternal age, season of conception, and on gestational week of first infection (i.e. the control pregnancy had to have reached at least the same gestation at that at which the infected pregnancy was first infected). We agree that it would have been ideal to match on exact date of conception (rather than the broader season of conception), but unfortunately this was not feasible due to the limited pool of controls available. We therefore prioritised exact matching on maternal age and gestation at infection/matching, with looser matching on season of conception. We have revised our Discussion section to be clear that this is a limitation to the analysis:

“We were also unable to match pregnancies on exact calendar time (i.e., matching exposed and control babies by the calendar week of estimated conception) due to an insufficient pool of controls, but were able to match on season of conception for our infection analyses.”

Line 205: Very important. It is unclear why conditional logistic regression was used for primary analyses when this is a matched cohort study. Other matched modeling methods, such as conditional Poisson, are available and would provide the appropriate measure (risk ratios) for a matched cohort study instead of odds ratios, which may overestimate the effect.

Author response: While the reviewer is right that risk ratios are often easier to interpret, they are much harder to statistically model (models often do not converge). If our outcomes were common, then we would agree that it would be worth exploring using others models (although, given that our outcomes are binary rather than time to event or counts, we would argue that Poisson would not be the appropriate method). However, as our outcomes are generally rare, the odds ratios will give a very close approximation to the risk ratios and negates the need for more complex modelling.

We gave the analytical approach extensive consideration in developing our protocol and did specifically consider the use of time-to-event/survival analysis methods. For the majority of our outcomes, however, we felt that it would be inappropriate to use these methods for the following reasons:

1. For the majority of our baby outcomes, it is unnecessary to use survival analysis methods as these outcomes are captured specifically around the time of birth and so timing of the event is fixed (e.g. small-for-gestational age, Apgar score, and stillbirth).
2. For the majority of our maternal outcomes (pregnancy-related bleeding/pregnancy-related hypertension/venous thromboembolism), we do not have a reliable incident date for outcomes that occurred in the antenatal period as antenatal complications may only be recorded on the delivery record.

Additionally, we have a relatively short follow up period for cohort members (to the of the neonatal or postpartum period) and no loss to follow up, hence the benefits of survival analysis in terms of dealing with differential follow up and censoring do not apply. Our current approach still allowed us to take advantage of the longitudinal nature of the data, in that we were able to establish the timing of the exposure in relation to the outcome in a robust way.

In the results and discussion sections:

Very nice Table 1. Helpful to the reader.

Author response: Thank you.

Table 3: Consider noting reference groups explicitly where model estimates are provided (e.g., "Ref" instead of "1")

Author response: We have now revised our tables to show the reference groups with “Ref” rather than 1.

Figure 1: Could the authors consider using alternatively white/gray shaded rows in their forest plot to make the figure a little bit easier to read? (This can be done fairly quickly in the forest package in R.)

Author response: Thank you for this suggestion. We have now revised Figure 1 to include alternating white/grey shaded areas for each different outcome and think this improves the readability of the graph.

Line 209: Did the authors consider imputing missing data, and comparing complete-case to imputed results to assess bias?

Author response: We assume the reviewer is referring to missing data for the covariates, and we did consider several potential options for dealing with these, including imputation. We ultimately decided this was unlikely to be an appropriate approach to deal with our missing covariate data for two reasons:

1. For covariates where data were likely to be missing completely at random (e.g., maternal urban rural status and maternal deprivation), there were very low levels of missing data so any imputation would have negligible impact.
2. For the few covariates with higher levels of missing data, these were unlikely to be missing completely at random (e.g. maternal ethnicity) and we instead included an unknown group.

As noted in the manuscript, we conducted a sensitivity analysis to assess the impact of the missing data, specifically conducting a complete case analyses dropping any pregnancies with missing covariate information to reassure that this was not biasing our results and found negligible changes to our results.

Line 225: Please specify "weeks" when discussing before or after 20 weeks of gestation.

Author response: We have now revised this sentence to read:

“Pregnancies where the woman had infections at < and ≥ 20+0 weeks gestation were excluded.”

Line 298: Suspect incorrect date. Did you intend to state that your COVID-19 vaccination program began on December 8, 2021?

Author response: Thank you for spotting this typo. We have now revised this to December 8, 2020, which was the start of the vaccination programme.

Lines 341-343: As Allotey et al. are continuously updating their findings, you could probably withhold providing the OR and 95%CI estimates and simply state the outcomes that infection during pregnancy are associated with.

Author response: We think it is helpful to give readers a sense of how similar the magnitude of our odds ratios are compared with what has been reported in the literature, but agree with the reviewer that these may end up out of sync with any updated findings from Allotey et al. We have therefore added to the description of these results the cut-off for the end of the iteration of the systematic review, which we report in the Discussion as follows (addition underlined):

“Our results align with those of a living systematic review and meta-analysis that found, based on literature identified up to 27th April 2021, that pregnant women with SARS-CoV-2 infection had increased odds of maternal death (OR=6.09, 95% CI=1.82-20.38), admission to the intensive care unit (OR=5.41, 95% CI=3.59-8.14), preterm birth (OR=1.57, 95% CI=1.36-1.81), stillbirth (OR=1.81, 95% CI=1.38-2.37), and admission to the neonatal intensive care unit (OR=2.18, 95% CI=1.46-3.26).”

Line 421: Suggest modifying to state "no clear need for enhanced ongoing pregnancy monitoring due to infection in early pregnancy." (Essentially, to imply that antenatal care is still vitally important.)

Author response: We have revised this sentence as suggested by the reviewer.

Reviewer #3 (Remarks to the Author):

Thank you for the opportunity to review this manuscript. This is a national study from Scotland on the impact of SARS-CoV-2 infection and COVID-19 vaccination on infant and maternal outcomes. Infection was associated with preterm birth, maternal admission to critical care or death and VTE. There was no increased risk associated with vaccination against COVID-19. The study is of high quality using national Scottish data.

General comments

1. It is not clear what proportion of women were tested for SARS-CoV-2 infection and when during pregnancy this was performed. From previous studies we know that many women who tested positive at delivery did not have symptoms of COVID-19. Were only women with symptoms tested? Were all women tested on admission to delivery? Was there information also on negative tests?

Author response: Thank you for raising this important point. Within the COPS study database we have access to all positive RT-PCR tests for SARS-CoV-2 and, from 6 Jan 2022, also positive LFD tests that were not followed by a negative RT-PCR within 48 hours. This allows us to identify all virologically confirmed infections in line with Scottish Government guidance. The test results are from all laboratories across Scotland, and hence reflect positive tests taken in hospital, in community testing centres, and (for LFD tests) at home, either in response to symptoms/clinically suspected infection or as part of regular/routine testing of asymptomatic individuals.

In common with other countries, testing policy in Scotland evolved throughout the pandemic. At the start of the pandemic, RT-PCR testing was restricted to individuals admitted to hospital, and healthcare staff, with clinically suspected COVID-19. From 18 May 2020, RT-PCR testing was also made freely available on request to all individuals across Scotland with relevant symptoms through a network of community testing centres. In addition to this symptom-driven testing, RT-PCR testing for asymptomatic individuals was also gradually rolled out. For example, regular testing for health and social care staff (a group which will include some pregnant women) was implemented from summer 2020, and routine testing of all individuals admitted on a planned or emergency basis to hospital for any reason (including maternity care) was implemented from December 2020. LFD testing for asymptomatic individuals was also increasingly available from the end of 2020, with regular testing at home recommended for all individuals in the population from April 2021, supported by free provision of LFD tests through community pharmacies (see sequential national testing strategies published here and here). Until 5 Jan 2022, individuals with a positive LFD test were asked to undertake a subsequent RT-PCR test to confirm infection. From 6 Jan 2022 onwards, a positive LFD test (not followed by a negative RT-PCR test within 48 hours) was taken as indicating a confirmed infection, reflecting the very high levels of infection in the community at that time following the spread of the omicron variant and hence the high positive predictive value of a positive LFD test.

For our infection analysis, we restricted our study period to the date from which widespread community testing for symptomatic individuals was implemented (18 May 2020), to minimise the impact of the substantial under-ascertainment of symptomatic infections associated with restricted testing prior to that point.

The confirmed infections ascertained in our study from that point on will include symptomatic infections and some asymptomatic infections picked up in regular/routine testing. As noted in our Discussion, we do not have information on symptom status associated with test results, hence we cannot provide exact information on this split. Surveillance data on all virologically confirmed

infections in pregnant women published by Public Health Scotland (and based on analysis of data in the COPS study database) has previously shown that confirmed infections in pregnant women have been broadly evenly distributed across the three trimesters, reflecting the widespread access to testing throughout pregnancy for all women. We do not have access to negative test results within the COPS study database, hence cannot provide information on overall testing rates.

We have extended the relevant section of the Methods – Exposures section as follows (additions underlined):

“National data on confirmed SARS-CoV-2 infections and COVID-19 vaccinations were linked to the COPS database to identify exposures of interest. For infection analyses, we identified confirmed infections by a positive SARS-CoV-2 reverse transcription polymerase chain reaction (RT-PCR) test or, from Jan 6, 2022, a positive lateral flow device (LFD) test (unless the LFD result was followed by a negative RT-PCR result within 48 hours)¹⁹. Tests taken in hospital, in community testing centres, and (for LFD tests) at home, in response to symptoms or as part of regular/routine testing of asymptomatic individuals, were all included. A subsequent positive test during the relevant exposure period was considered a separate infection if it was >90 days after a prior infection.”

We have also extended the relevant section of the Discussion as follows (additions underlined):

“We relied on the results of viral testing to identify confirmed SARS-CoV-2 infections. Testing policy in Scotland evolved throughout the pandemic^{32,33}. For our infection analyses, we restricted our study period to the date from which widespread community testing for symptomatic individuals was implemented (18 May 2020), to minimise the impact of the substantial under-ascertainment of symptomatic infections associated with restricted testing prior to that point. It is still possible however that we may have under-ascertained some (generally mild) symptomatic infections if individuals did not get tested. In addition to testing of individuals with symptoms, routine testing of asymptomatic individuals was made increasingly available as the pandemic progressed, for example routine testing of all individuals admitted to hospital was implemented from December 2020. It is likely therefore that the confirmed infections included in our study will include both symptomatic and asymptomatic infections. We did not have reliable data on symptoms or severity of infection however, so we could not provide information on the relative numbers or explore how association with adverse outcomes varied by symptom status. Also, we did not have access to negative test results within the COPS study database, hence could not provide information on overall testing rates. We may therefore have misclassified some women with asymptomatic or mild infection as uninfected if they did not get tested, although we did restrict to the period when widespread community testing was available to minimise this.”

2. Was it possible to address the various variants of SARS-CoV-2 viruses during the study period. Research has found different results based on the variants. It is stated that the study matched for season of conception but did this also include the variants of the virus?

Author response: We have previously published data on maternal and baby outcomes following SARS-CoV-2 infection in pregnancy in different time periods when different viral variants were dominant in Scotland, and agree with the reviewer that understanding potential differential effects by variant is important, but did not include such analyses in this study for two main reasons:

1. We do not have reliable data on viral variants in the COPS study.
2. We would have low power to look at the rarer outcomes if we start to stratify by time period of infection (as a proxy indicator of variant), which is what would be required to try and tease this out.

This mentioned in the limitations section of the Discussion as follows:

“We did not have reliable data on ~~symptoms or severity of infection, nor on the viral variant~~ so we could not explore whether the associations varied by variant ~~these factors~~. The infections in pregnancy that are included in our analyses occurred over a relatively wide time period, from May 2020 to February 2022. It is therefore likely that included infections were caused by a range of viral variants, from wild type through Alpha, Delta and Omicron as these were sequentially dominant in Scotland across this time frame. We have previously shown that infections in the period when Omicron (B.1.1.529) was the dominant variant were associated with lower risk of adverse baby and maternal outcomes compared with infections when Delta (B.1.617.2) was the dominant variant³⁴.”

3. It is likely that women with signs of preterm birth were more often tested and this could influence the findings. Hence, in parallel with comment no 1, testing strategy is likely to affect the results.

Author response: We recognise that it is possible that women with signs of preterm birth were more often tested and this could influence the findings, and such potential ascertainment bias is discussed as a limitation. However, we think this is unlikely to have significantly affected our results for the following reasons:

1. As noted in our response to the reviewer’s first point above, testing was freely available to all individuals with symptoms suggestive of SARS-CoV-2 infection throughout our study period. In addition, routine testing of all individuals admitted to hospital for any reason was also implemented during our study period, from December 2020 onwards. This will have included testing of women admitted for delivery at any gestation.
2. There was no associated increase in some other pregnancy complications that are associated with increased hospital admissions. For example, hypertensive disorders of pregnancy increase attendance and admission to hospital but we found no association between infection and increased risk of this outcome.

Specific comments

1. It is unclear why pregnancies conceived towards the end of the inclusion period were retained. Could this not lead to a selection of pregnancies and hence influence the results?

Author response: When developing our study protocol, we specified that we would include pregnancies conceived up to 1 June 2021. This would allow all included pregnancies to be followed up to 40+6 weeks gestation by end February 2022, therefore allowing sufficient time for the subsequent return of records so both the maternal and baby outcomes could be measured. In practice, as we included pregnancies delivering up to 44+6 weeks gestation, a very small number of included pregnancies conceived shortly before 1 June 2021 were not delivered until early March 2022. There was still time for relevant records to be returned on these pregnancies prior to data extraction however, hence no selection bias was introduced. This is described in the Methods - Setting and participants section:

“To allow sufficient time for the return and incorporation of records relating to the end of pregnancy plus the four-week neonatal (for the baby) or six-week postpartum (for the mother) period prior to data extraction, we excluded pregnancies conceived after June 1, 2021, to ensure all included pregnancies could be followed-up to the likely upper gestational limit of 40+6 gestation by February 28, 2022. However, to minimise unnecessary loss of data, pregnancies conceived towards the end of the inclusion period were retained even if they delivered after 40+6 (up to 44+6) and hence a small number of pregnancies ended in March 2022.”

2. The adjusted OR for hypertensive disease of pregnancy was 0.82 (0.68-0.98). Still it is stated that infection was not associated with hypertensive disease of pregnancy? Please revise and comment on this finding.

Author responses: We found no evidence for an association between infection and an increased risk of hypertensive disorders of pregnancy, and have tried to be very clear in our interpretation of our results:

*“We found no evidence for an association between infection and an **increased risk** of hypertensive disorders of pregnancy or pregnancy-related bleeding.”*

We considered the finding of the apparent association between infection and reduced risk of hypertensive disorders of pregnancy at length as it was unexpected. There is no biologically plausible reason to think that infection would lead to a decreased risk of hypertensive disorder of pregnancy, hence we consider this specific result is most likely to be either due to chance or potentially residual confounding. We have not focussed on this in our Discussion as we do not find a signal of increased risk.

3. In the conclusion it is stated that increased risks of adverse outcomes are associated with infection in later, not earlier. This has been shown in previous studies and would hence not be stated as novel.

Author response: We have amended the relevant sentence as follows:

“We also ~~provide novel findings showing~~ show that the increased risks are associated with infections in later, but not earlier, pregnancy.”

REVIEWER COMMENTS

Reviewer #1 (Remarks to the Author):

The authors addressed all my comments with satisfaction. I have no additional comments. Great job.

Reviewer #2 (Remarks to the Author):

I appreciate the authors' time and effort spent addressing reviewers' concerns. The authors adequately addressed some, but not all, of the issues raised. This study does have merit and could be a valuable contribution to the literature, but the following issues need to be addressed:

Terminology: I strongly encourage the use of the terms "infant" or "neonate," as "baby" is considered too colloquial and is no more specific than "infant" or "neonate." "Infant" or "neonate" are the terms more commonly used in medical literature. Infants or neonates can still be stillborn (the authors can define them as "stillborn infants.") In addition, just as the term "infant" includes those beyond 28 days of age, so does the term "baby."

The terms "neonate" or "neonatal outcomes" within the first 28 days of age may be even better. Could the authors provide a reference that indicates that "infant" or "neonate" applies only to those who are liveborn, as I have not seen this? Or that "baby" does not include those beyond 28 days of age?

Methods: Please also modify the methods to explicitly state that infants are not followed beyond 28 days. All inclusion and exclusion criteria should be defined very early and clearly in the "Setting and Participants" paragraph, directly indicating that infants were only followed for their first 28 days; however, in this paragraph, there was only a passing reference to this inclusion criterion: Lines 116-118 ("To allow sufficient time for the return and incorporation of records relating to the end of pregnancy plus the four-week neonatal (for the baby)... prior to data extraction.")

Modeling: While the outcomes are generally rare (indicating that ORs will likely approximate RRs), the data should still be analyzed in the manner in which they were collected. The authors are correct that log-binomial methods are subject to convergence issues; however, Poisson regression is a perfectly suitable and established approach to calculate risk ratios. The authors suggest that Poisson models are not appropriate for binary outcomes, but Poisson models are commonly used for binary outcome data in cohort studies, not just count data. They are advantageous particularly where data are sparse or outcomes are rare, as they avoid model convergence issues that are more common with log-binomial modeling. As such, reconsideration of modeling methods is needed. Please see the following references:

Chen, W., Qian, L., Shi, J. et al. Comparing performance between log-binomial and robust Poisson regression models for estimating risk ratios under model misspecification. *BMC Med Res Methodol* 18, 63 (2018). <https://doi.org/10.1186/s12874-018-0519-5>

Guangyong Zou, A Modified Poisson Regression Approach to Prospective Studies with Binary Data, *American Journal of Epidemiology*, Volume 159, Issue 7, 1 April 2004, Pages 702–706, <https://doi.org/10.1093/aje/kwh090>

I agree that survival analyses are not needed, but please reconsider your modeling approach to calculate risk ratios.

Reviewer #3 (Remarks to the Author):

My suggestions for revision of the manuscript have been addressed and I have no additional comments.

RESPONSE TO REVIEWER COMMENTS

Reviewer #1 (Remarks to the Author):

The authors addressed all my comments with satisfaction. I have no additional comments. Great job.

Author response: Many thanks.

Reviewer #2 (Remarks to the Author):

I appreciate the authors' time and effort spent addressing reviewers' concerns. The authors adequately addressed some, but not all, of the issues raised. This study does have merit and could be a valuable contribution to the literature, but the following issues need to be addressed:

Terminology: I strongly encourage the use of the terms “infant” or “neonate,” as “baby” is considered too colloquial and is no more specific than “infant” or “neonate.” “Infant” or “neonate” are the terms more commonly used in medical literature. Infants or neonates can still be stillborn (the authors can define them as “stillborn infants.”) In addition, just as the term “infant” includes those beyond 28 days of age, so does the term “baby.”

The terms “neonate” or “neonatal outcomes” within the first 28 days of age may be even better. Could the authors provide a reference that indicates that “infant” or “neonate” applies only to those who are liveborn, as I have not seen this? Or that “baby” does not include those beyond 28 days of age?

Author response: We acknowledge the reviewer’s views here and agree that neonate/neonatal is more appropriate for our manuscript than infant. We have therefore revised the manuscript, replacing baby/babies with neonate/neonatal through the paper. Our title, for example, is now: *“Neonatal and maternal outcomes following SARS-CoV-2 infection and COVID-19 vaccination during pregnancy: a national population-based matched cohort study”*

We have also added in clarification of the use of this terminology in the **“Setting and Participants”** section of the Methods as follows (addition underlined):

“For this study, we included singleton pregnancies that reached at least 20 weeks, 0 days (20+0) gestation and ended before 44 weeks, 6 days (44+6) gestation in a live or stillbirth. Hereafter, we use the term “neonate” as the overarching term including these live and stillbirths.”

Methods: Please also modify the methods to explicitly state that infants are not followed beyond 28 days. All inclusion and exclusion criteria should be defined very early and clearly in the “Setting and Participants” paragraph, directly indicating that infants were only followed for their first 28 days; however, in this paragraph, there was only a passing reference to this inclusion criterion: Lines 116-118 (“To allow sufficient time for the return and incorporation of records relating to the end of pregnancy plus the four-week neonatal (for the baby)... prior to data extraction.”)

Author response: Thank you for this suggestion. We have now added the following sentence to our **“Setting and Participants”** section in the Methods:

“Live born neonates were followed up to four weeks (28 days) after birth and all mothers up to six weeks (42 days) postpartum.”

Modeling: While the outcomes are generally rare (indicating that ORs will likely approximate RRs), the data should still be analyzed in the manner in which they were collected. The authors are correct that log-binomial methods are subject to convergence issues; however, Poisson regression is a

perfectly suitable and established approach to calculate risk ratios. The authors suggest that Poisson models are not appropriate for binary outcomes, but Poisson models are commonly used for binary outcome data in cohort studies, not just count data. They are advantageous particularly where data are sparse or outcomes are rare, as they avoid model convergence issues that are more common with log-binomial modeling. As such, reconsideration of modeling methods is needed. Please see the following references:

Chen, W., Qian, L., Shi, J. et al. Comparing performance between log-binomial and robust Poisson regression models for estimating risk ratios under model misspecification. *BMC Med Res Methodol* 18, 63 (2018). <https://doi.org/10.1186/s12874-018-0519-5>

Guangyong Zou, A Modified Poisson Regression Approach to Prospective Studies with Binary Data, *American Journal of Epidemiology*, Volume 159, Issue 7, 1 April 2004, Pages 702–706, <https://doi.org/10.1093/aje/kwh090>

I agree that survival analyses are not needed, but please reconsider your modeling approach to calculate risk ratios.

Author response: Thank you for providing these helpful references.

We have re-analysed our data using conditional Poisson regression and are reassured that there are negligible differences between the resulting risk ratios and the odds ratios produced using conditional logistic regression. There is no difference in overall interpretation of results using the two approaches.

We have reflected on how best to integrate these new results. The conditional Poisson regression modelling represents a deviation from our pre-specified statistical analysis plan outlined in the study protocol. It reduces alignment of this paper with the two previous papers published in the same journal by our group that examine associations between SARS-CoV-2 infection and COVID-19 vaccination and early pregnancy loss (<https://www.nature.com/articles/s41467-022-33937-y>) and congenital anomalies (<https://www.nature.com/articles/s41467-022-35771-8>). We have therefore provided the results from the conditional Poisson regression models as additional supplementary material (Supplementary Table 6 for the infection analyses and Supplementary Table 11 for the vaccination analyses) so that readers can see the results using both approaches. We have noted in the Methods that we undertook these additional analyses to calculate risk ratios in a deviation from our protocol with the addition of the following sentence in the statistical analysis section:

“In response to reviewer request, in a deviation from our protocol we replicated these analyses using conditional Poisson regression allowing us to calculate risk ratios for comparison with the odds ratios from the conditional logistic regression models.”

We have signposted Supplementary Table 6 in the **“SARS-CoV-2 infection and neonatal outcomes”** section of the Results with the addition of the following sentence:

“As shown in Supplementary Table 6, risk ratios calculated using conditional Poisson regression showed negligible difference from the odd ratios calculated using conditional logistic regression.”

We have added a similar sentence to the **“SARS-CoV-2 infection and maternal outcomes”** section of the Results.

We have signposted Supplementary Table 11 in the **“COVID-19 vaccination and neonatal and maternal outcomes”** section of the Results with the following edit (addition underlined):

“After adjusting for covariates, we found no evidence that maternal COVID-19 vaccination was associated with increased risk of any of the neonatal or maternal outcomes (Table 4 &

Supplementary Table 10), and there was negligible difference between the risk ratios and the odds ratios (Supplementary Table 11).

Reviewer #3 (Remarks to the Author):

My suggestions for revision of the manuscript have been addressed and I have no additional comments.

Author response: Many thanks.